# Spatially-resolved transcriptomics reveal macrophage heterogeneity and prognostic significance in diffuse large B-cell lymphoma

Min Liu[1,2,3], Giorgio Bertolazzi[4,5], Shruti Sridhar[1], Rui Xue Lee[1], Patrick Jaynes[1], Kevin Mulder [6,7,8], Nicholas Syn [9,10], Michal Marek Hoppe [1], Shuangyi Fan[9], Yanfen Peng [1], Jocelyn Thng[1], Reiya Chua[11], Jayalakshmi[11], Yogeshini Batumalai[11], Sanjay De Mel [11,12], Limei Poon[11,12], Esther Hian Li Chan[11,12], Joanne Lee[11,12], Susan Swee-Shan Hue[9,12], Sheng-Tsung Chang[13], Shih-Sung Chuang[13], K. George Chandy[14], Xiaofei Ye [15], Qiang Pan-Hammarström [16], Florent Ginhoux [6,7,8], Yen Lin Chee[11,12], Siok-Bian Ng [1,9,12], Claudio Tripodo [5,17] ✉ & Anand D. Jeyasekharan [1,11,12,18] ✉

Macrophages are abundant immune cells in the microenvironment of diffuse large B-cell lymphoma (DLBCL). Macrophage estimation by immunohistochemistry shows varying prognostic significance across studies in DLBCL, and does not provide a comprehensive analysis of macrophage subtypes. Here, using digital spatial profiling with whole transcriptome analysis of CD68+ cells, we characterize macrophages in distinct spatial niches of reactive lymphoid tissues (RLTs) and DLBCL. We reveal transcriptomic differences between macrophages within RLTs (light zone /dark zone, germinal center/ interfollicular), and between disease states (RLTs/ DLBCL), which we then use to generate six spatially-derived macrophage signatures (MacroSigs). We proceed to interrogate these MacroSigs in macrophage and DLBCL single-cell RNA-sequencing datasets, and in gene-expression data from multiple DLBCL cohorts. We show that specific MacroSigs are associated with cell-of-origin subtypes and overall survival in DLBCL. This study provides a spatially-resolved whole-transcriptome atlas of macrophages in reactive and malignant lymphoid tissues, showing biological and clinical significance.

Diffuse large B-cell lymphoma (DLBCL) is the most common subtype of non-Hodgkin lymphoma in adults[1,2]. Combination therapy of rituximab with cyclophosphamide, doxorubicin, vincristine, and prednisolone (R-CHOP) is potentially curative, but 30%–40% of cases relapse after initial therapy[3]. Elucidating mechanisms underlying relapse following R-CHOP is critical for the development of therapeutic strategies to improve DLBCL outcomes. An important and emerging area is the role of the tumor microenvironment (TME) in mediating disease progression and clearance of tumor cells after chemotherapy[4–8]. Understanding factors in the TME that mediate DLBCL relapse following R-CHOP is also

important in terms of incorporating current immunotherapeutics such as bispecific CD20-CD3 T-cell engagers, CD19 CAR-T cells and anti-CD47 antibodies into front-line treatment regimens for DLBCL.

Tumor-associated macrophages (TAMs) are abundant immune cells in the DLBCL TME[9,10]. TAMs are recognized as potential therapeutic targets in oncology due to their role in tumor progression, metastasis, and recurrence[11]. In DLBCL, TAM infiltration is associated with poor prognosis after R-CHOP therapy[9,12,13]. However, there are discrepancies observed between studies, with a lack of sufficient reproducibility to identify consistent clinical prognostic markers[14]. These discrepancies

may partly result from the simplified functional classification of macrophages into M1/M2 phenotypes[15]. Conventionally, M1 macrophages refer to a pro-inflammatory phenotype with pathogen-killing abilities while M2 refers to an anti-inflammatory phenotype promoting proliferation, tissue repair and tumorigenesis[16]. However, macrophages in several physiological or pathological settings, including embryonic macrophages, resolution-phase macrophages and even certain TAMs do not clearly fall into either the M1 or M2 phenotype[17]. Furthermore, macrophages show high phenotypic plasticity[17], suggesting that the M1/M2 dichotomy does not fully encompass the functional diversity of macrophages[17–19]. Indeed, with single-cell transcriptomic approaches, the functional complexity of macrophages is now widely appreciated[20,21]. The nature of TAMs defined by comprehensive transcriptomic approaches in DLBCL and their relationship with treatment outcomes remain poorly understood[22].

In this work, we comprehensively characterize TAMs in DLBCL and reactive lymphoid tissues (RLTs) using digital spatial profiling (DSP), an advanced technique for spatially resolved transcriptomics. The DSP whole transcriptome atlas (WTA) provides an unbiased map of 18,000 + RNA targets throughout specific cell types of interest (chosen based on a fluorescent protein marker- here CD68 for macrophages) in formalin-fixed paraffin-embedded tissue sections. Here, using DSP, we generate spatially-derived macrophage transcriptomic signatures and explore their associations with previously described macrophage subpopulations and clinical/ biological features of DLBCL.

## Results

### Digital spatial profiling (DSP) illuminates consistent profiles from distinct masks in lymphoid tissue microregions

We profiled the whole transcriptome of macrophages, T cells, and B cells using the GeoMx® DSP WTA assay (Fig. 1A), by selective collection of UV-cleavable probes from distinct masks generated by immunofluorescent staining of the morphology markers CD68, CD3 and CD20 (pertaining to macrophages, T-cells and B-cells, respectively), in RLTs ($n = 24$) and DLBCL patients ($n = 64$) for a total of 702 areas of interest (AOIs) (Supplementary Fig. 1A, representative images in Fig. 1B, C and Supplementary Fig. 1B–D).

Given the heterogeneity and high cellularity of the DLBCL microenvironment, we aimed at confirming that DSP-based cell selection generated reliable cell-type specific profiles. We therefore cross-validated our DSP output with publicly available scRNA-seq data of RLTs and DLBCL specimens. To do this, we defined gene signatures characteristic of macrophages, T cells and B cells using 4–5 hallmark genes of these cell types (Supplementary Table 1). As expected, these signatures projected to matching cell-types in scRNA-seq datasets of RLTs (Supplementary Fig. 2) and DLBCL (Supplementary Fig. 3). We then tested these signatures on our DSP based CD68+ (macrophage), CD20+ (B-cell) and CD3+ (T-cell) AOIs using cumulative density functions. We show that the macrophage signature was enriched in CD68+ AOIs, while the signatures of T and B cells were enriched in CD3+ and CD20+ AOIs, respectively (adjusted $P < 0.05$, Fig. 1D, E). This suggests that morphology marker-based AOIs accurately capture the respective cell types of interest, enabling the collection of distinct whole transcriptome profiles of macrophages, T cells and B cells in their native tissue environment.

We also verified the robustness of our DSP experiment by cross-comparing its output for differentially expressed genes (DEGs) between the light zone (LZ) and dark zone (DZ) regions (all cells, majority CD20 + ) in the germinal center (GC) of RLTs with previously published information on transcriptional differences between the LZ and DZ[23] (Supplementary Fig. 4A, B and Supplementary Data 1). Almost all DEGs of DZ and LZ regions from our DSP experiment overlapped with previously reported LZ/DZ DEGs from ref. 23 (Supplementary Fig. 4C), further confirming the reliability and adequate transcriptomic coverage of our DSP approach for subsequent analyses and inferences.

### Unique gene expression patterns differentiate macrophages in distinct spatial locations within reactive lymphoid tissues

Having quality checked the DSP data, we next aimed to investigate transcriptomic differences of CD68+ macrophages within different spatial regions of RLTs. Through DEG analysis, we observed that 997 and 755 genes were differentially upregulated in the GC and IF macrophages, respectively (adjusted $P < 0.05$ and $|\log_2 FC| > 0.58$, Fig. 2A), suggesting highly distinct gene expression patterns. The heatmap in Fig. 2B displays the top 10 DEGs highly expressed in the GC and IF regions. Cell proliferation and metabolism-associated pathways such as E2F targets, MYC targets, and oxidative phosphorylation were enriched pathways in GC macrophages (adjusted $P < 0.0001$, Fig. 2C). These proliferative pathways such as E2F transcription factors are critical for a wide variety of cell-types including myeloid progenitors and differentiated macrophages[24]. In contrast, enriched pathways in IF were mostly associated with immune responses such as the interferon γ response and TNF-α/NF-κB pathways (adjusted $P < 0.0001$, Fig. 2C). Of interest, macrophages in the IF showed upregulation of S100A family members (calcium binding proteins), such as *S100A4*, *S100A8*, and *S100A9*, which are known Damage-Associated Molecular Pattern (DAMP) molecules regulating macrophage biology[25] (Supplementary Data 2).

The GC consists of two functionally distinct compartments: dark zone (DZ) and light zone (LZ)[26,27]. This compartmentalization is critical for dynamic differentiation of B cells within the GC[28,29]. We also compared the DEGs of macrophages in the LZ and DZ. We also note significant differences (adjusted $P < 0.05$ and $|\log_2 FC| > 0.58$, Fig. 2D) in gene expression between macrophages populating these anatomically distinct compartments of the germinal center. The complement pattern recognition components *C1QA*, *C1QB*, and *C1QC* were significantly upregulated in DZ macrophages (adjusted $P < 0.0001$, Fig. 2D), as were other components of the Hallmark gene set of the complement pathway (Fig. 2E). This suggests a possible role for non-canonical complement system functions in macrophage polarization in the DZ of RLTs.

Based on the above comparisons between macrophages in distinct spatial locations, we derived macrophage signatures (termed henceforth as MacroSigs) from the respective DEGs meeting the following criteria: a. Benjamini-Hochberg adjusted $P < 0.05$; b. $|\log_2 FC| > 0.58$[30–32]. MacroSigs (generated from upregulated DEGs) corresponding to spatial compartments in RLTs were: MacroSig1 (GC), MacroSig2 (IF), MacroSig3 (LZ), and MacroSig4 (DZ). We then evaluated if these spatially-derived MacroSigs could be mapped to known macrophage subclusters generated through single-cell RNA sequencing. We utilized an integrative dataset named MoMac-VERSE[33] (Fig. 2F), the current largest meta-analysis of human monocytes and macrophages, with 17 annotated monocyte/macrophage subclusters from 41 scRNA-seq datasets comprising 13 healthy and pathological tissues. We projected the top 50 genes of our MacroSigs (Supplementary Data 2) onto MoMac-VERSE and noted that MacroSig1 (GC) overlapped with TREM2+ macrophages (Fig. 2G). Interestingly, despite having clear transcriptional distinctions from GC macrophages, the MacroSig2 (IF) was dispersed and did not overlay with one or more specific subclusters of macrophages as defined by the MoMac-VERSE (Fig. 2H). This raised the possibility that MacroSigs from distinct regions of lymphoid tissue may denote hitherto unknown macrophage subtypes (not represented in the MoMac-VERSE metanalysis, which is not a spatially resolved approach). Additionally, MacroSig3 (LZ) localized to the MNP/T cell doublets (Fig. 2I), while MacroSig4 (DZ) overlapped with the HES1/FOLR2 macrophage population (Fig. 2J). These results indicated that these macrophage subpopulations may play specific roles in distinct regions of lymphoid tissues, deserving functional investigation.

### Distinct transcriptomic profiles of macrophages between reactive and malignant lymphoid tissue

We next compared the gene expression of macrophages from RLT germinal center (GC) regions with that of macrophages from

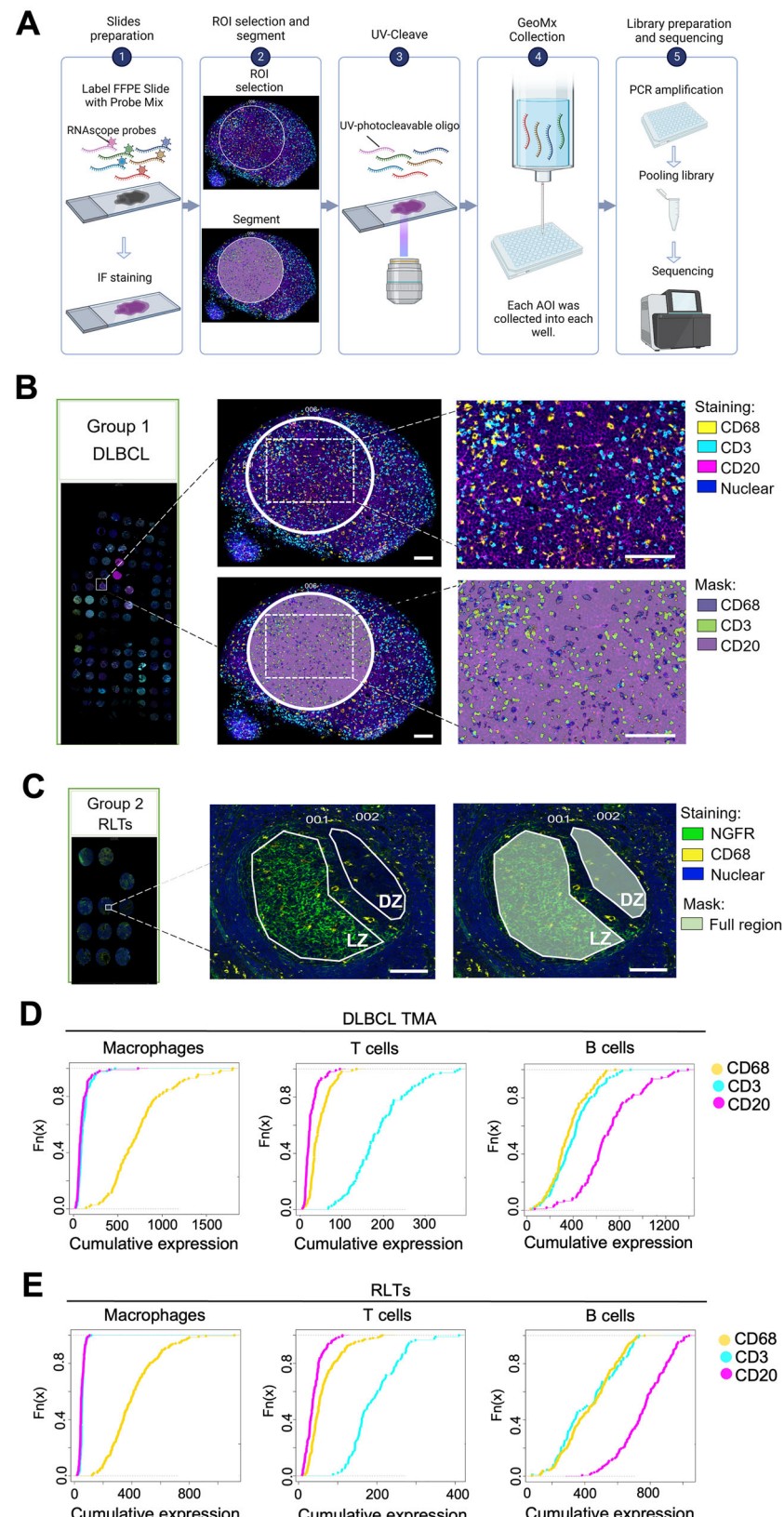

DLBCL samples. The DEGs of these two macrophage subsets highlight that 895 and 468 unique genes were differentially upregulated in RLTs GC and DLBCL macrophages, respectively (Fig. 3A). The heatmap in Fig. 3B displays the top DEGs highly expressed in both RLT and DLBCL. These DEGs are referred to henceforth as MacroSig5 (RLT) and MacroSig6 (DLBCL) using the above-mentioned criteria (adjusted

$P < 0.05$ and $|\log_2 FC| > 0.58$). As the features of macrophages in distinct spatial locations of RLTs have been detailed in the previous section, here we focus on features of macrophages in DLBCL. We noted that *CD163*, a marker of pro-tumorigenic macrophages, as well as complement pattern recognition components (*C1QA*, *C1QB*, and *C1QC*) were markedly upregulated in DLBCL

**Fig. 1 | Digital spatial profiling (DSP) illuminates consistent profiles from distinct masks in lymphoid tissue microregions. A** Schematic of GeoMx® DSP WTA workflow (created with BioRender.com). **B**, **C** Immunofluorescence staining of DLBCL tissues (n = 87) and RLTs (n = 24). In Group 1, CD68 stained macrophages (yellow), CD3 stained T cells (cyan), CD20 stained B cells (magenta), and SYTO 13 stained nuclei (blue). In Group 2, CD68 stained macrophages (yellow), NGFR illuminated LZ (green) and SYTO 13 stains nuclei (blue). After ROI selection, each cell type was segmented based on the staining signal and their corresponding masks were generated. Representative images are shown. Scale bar: 100 μm.

Source data are provided as a Source Data file. **D**, **E** Cumulative density functions showed that the signatures of macrophages (*CD68*, *CD163*, *FCGR1A*, and *CSF1R*), T cells (*CD3D*, *CD3E*, *UBASH3A*, *CD2*, and *TRBC2*), and B cells (*MS4A1*, *CD79A*, *CD79B*, *CD19*, and *PAX5*) were highly enriched in CD68+ regions, CD3+ regions, and CD20+ regions, respectively in RLTs and DLBCL tissues (Kolmogorov-Smirnov *P* < 0.05). Digital spatial profiling, DSP; whole transcriptome analysis, WTA; diffuse large B-cell lymphoma, DLBCL; reactive lymphoid tissues, RLTs; regions of interest, ROIs; areas of interest, AOIs; formalin-fixed paraffin-embedded, FFPE; light zone, LZ; dark zone, DZ; nerve growth factor receptor, NGFR.

---

macrophages (adjusted *P* < 0.0001, Supplementary Data 2). Enriched pathways in the DLBCL MacroSig were mostly associated with immune responses such as interferon response and complement pathways (adjusted *P* < 0.005, Fig. 3C). Using the MoMac-VERSE, we see that MacroSig6 (DLBCL) projected to the IL4I1+ macrophage population (Fig. 3D), a macrophage subset that embodies immunosuppressive functions in diverse cancer types[34–36]. These findings hint at a potential role of IL4I1+ macrophages in DLBCL pathogenesis and offer the prospect of exploring the targeting of these cells, although additional mechanistic work will be required to confirm the feasibility of such putative therapeutic interventions.

### Spatially-derived MacroSigs associate with COO DLBCL subclassifications

DLBCL patients are subtyped based on B cell-of-origin (COO) gene expression profiling (GEP). As such, we sought to understand if our spatially-derived MacroSigs were related to these subclassifications. To perform this analysis, we explored the enrichment of our MacroSigs in bulk RNA gene expression profiles of DLBCL patients across eight publicly available transcriptomic datasets (n = 4594, 8 datasets). Across all datasets, MacroSig1 (GC) was enriched in germinal center B-cell like (GCB) DLBCL (adjusted *P* < 0.05, Fig. 4A, 8/8 datasets), MacroSig2 (IF) in unclassified (UNC) DLBCL (adjusted *P* < 0.05, Figs. 4A, 6/8 datasets), and MacroSig6 (DLBCL) in activated B-cell (ABC) subtype DLBCL (adjusted *P* < 0.05, Fig. 4B, 8/8 datasets). MacroSig3 (LZ) and MacroSig4 (DZ) were not distinctly enriched in any COO category (Fig. 4C).

Given these interesting but unexpected correlations, we evaluated if the genes composing our MacroSigs identified subpopulations of macrophages in scRNA-seq data from DLBCL cases (n = 17, comprising transcriptomes of 94,324 cells)[37]. We see that our MacroSigs were enriched within distinct clusters of the monocyte-macrophage populations in the DLBCL scRNA-seq dataset (Fig. 4D and Supplementary Fig. 5). Most MacroSigs showed negligible module scores in the DLBCL B-cell population (except for GC/RLT [Fig. 4D and Supplementary Fig. 6], which share several cell-cycle/ proliferation genes with the GC/RLT B-cell signature from our DSP experiment). This analysis suggests that MacroSigs are indeed likely to represent macrophages in GEP analysis, even when evaluated in the malignant setting with aberrant gene expression profiles as compared with normal B cells. We further investigated if these MacroSigs could relate to recently established DLBCL genetic and molecular subtypes. Comparisons were conducted across 3 independent reference datasets[38–40]. Overall, we did not observe consistent enrichment of MacroSigs with specific genetic subtypes (Supplementary Fig. 7A). Similarly, there was no clear enrichment of our MacroSigs with specific DLBCL TME categories proposed by ref. 5 (Supplementary Fig. 7B). These results point to convergent mechanisms for tumor associated macrophage infiltration across broad genetic and TME subgroups.

### Spatially-derived MacroSigs stratify patients for survival in gene-expression profiling datasets of DLBCL

Finally, we aimed to evaluate if spatially-derived MacroSigs could have prognostic significance when evaluated in bulk gene expression data from clinical samples of DLBCL, using the above-mentioned eight

clinically annotated DLBCL datasets (n = 4594, 8 datasets). Cases enriched for MacroSig6 (DLBCL) had shorter OS than those with MacroSig5 (RLT) (Fig. 5A, adjusted *P* < 0.05, 6/8 datasets; hazard ratios (HR) and 95% confidence intervals (CI) for each dataset in Supplementary Table 2). This was corroborated in the Kaplan-Meier plots across six datasets (Fig. 5B–I, adjusted *P* < 0.05). It is possible that patients enriched in MacroSig5 (RLT) may represent those with less immunosuppressive TMEs and fewer TAMs than cases enriched for MacroSig6 (DLBCL). This result also highlights that the application of a macrophage derived gene signature may remain clinically relevant in bulk gene-expression data from these tumors. Of particular interest however, we observed that DLBCL patients with MacroSig4 (DZ) had significantly worse OS compared to those with MacroSig3 (LZ) across multiple DLBCL datasets (Fig. 6A-I, adjusted *P* < 0.05, 7/8 datasets; HR and 95% CI refer to Supplementary Table 2). This association was also validated in a multivariate analysis adjusted for the IPI score and double hit lymphoma cases (Supplementary Table 3, adjusted *P* < 0.05, 5/6 datasets; IPI scores not available in 2 datasets; DHL only available in 2 datasets), confirming that MacroSig4 (DZ) is prognostic factor independent of clinical high- risk features for survival of DLBCL patients.

### Additional evaluation of the Dark Zone MacroSig in DLBCL

Gene expression signatures of B-cell dark-zone biology are now well appreciated to be prognostic in DLBCL, overlapping with molecular high-grade and double-hit like signatures[41]. We therefore evaluated the prognostic impact of the dark zone signature of B-cells obtained from our own DSP experiments (Supplementary Data 3), to compare it with our results from the DZ-MacroSig4. The B-cell-based DZ signatures were indeed prognostic (Fig. 7A, adjusted *P* < 0.05, 4/8 datasets), but interestingly with less consistency than the DZ-MacroSigs (Fig. 6A, adjusted *P* < 0.05, 7/8 datasets). Of note, only few genes were shared between the B-cell derived and macrophage-derived LZ- and DZ- signatures (MacroSig3-4) (Fig. 7B), highlighting that DLBCLs carrying features of these distinct aspects of dark zone biology (B-cell and Macrophage) have adverse outcomes, but likely through distinct biological mechanisms.

To further validate the DZ signature, we aimed to perform a protein based multiplex staining approach. As complement pattern recognition component genes (*C1QA*, *C1QB*, and *C1QC*) appeared as top ranking genes in the DZ signature (MacroSig4, Supplementary Data 2), and since C1Q expressing macrophages are a defined entity[42], we evaluated the immunofluorescence staining of C1Q in RLTs and also in an additional independent set of DLBCL tissues in tissue microarray format from the Chi-Mei Medical Center (CMMC), Taiwan. Using a co-stain with the dark zone marker AID, we note that C1Q expressing macrophages were indeed enriched in the dark zone (DZ) when compared to the light zone (LZ) in germinal centers from RLTs (Fig. 7C). In the CMMC cohort of DLBCL cases, we observed that patients with high C1Q expressing macrophages had poorer survival compared to those with low C1Q in their macrophages (Log-rank *P* < 0.05, Fig. 7D, E). As C1Q was also a differentially expressed hit in the DLBCL MacroSig (which was also prognostic in DLBCL), additional comparative staining of proteins coded within the MacroSigs will be required to tease apart different subtypes of C1Q expressing macrophages and their relationship to the poor prognostic dark-zone B-cell

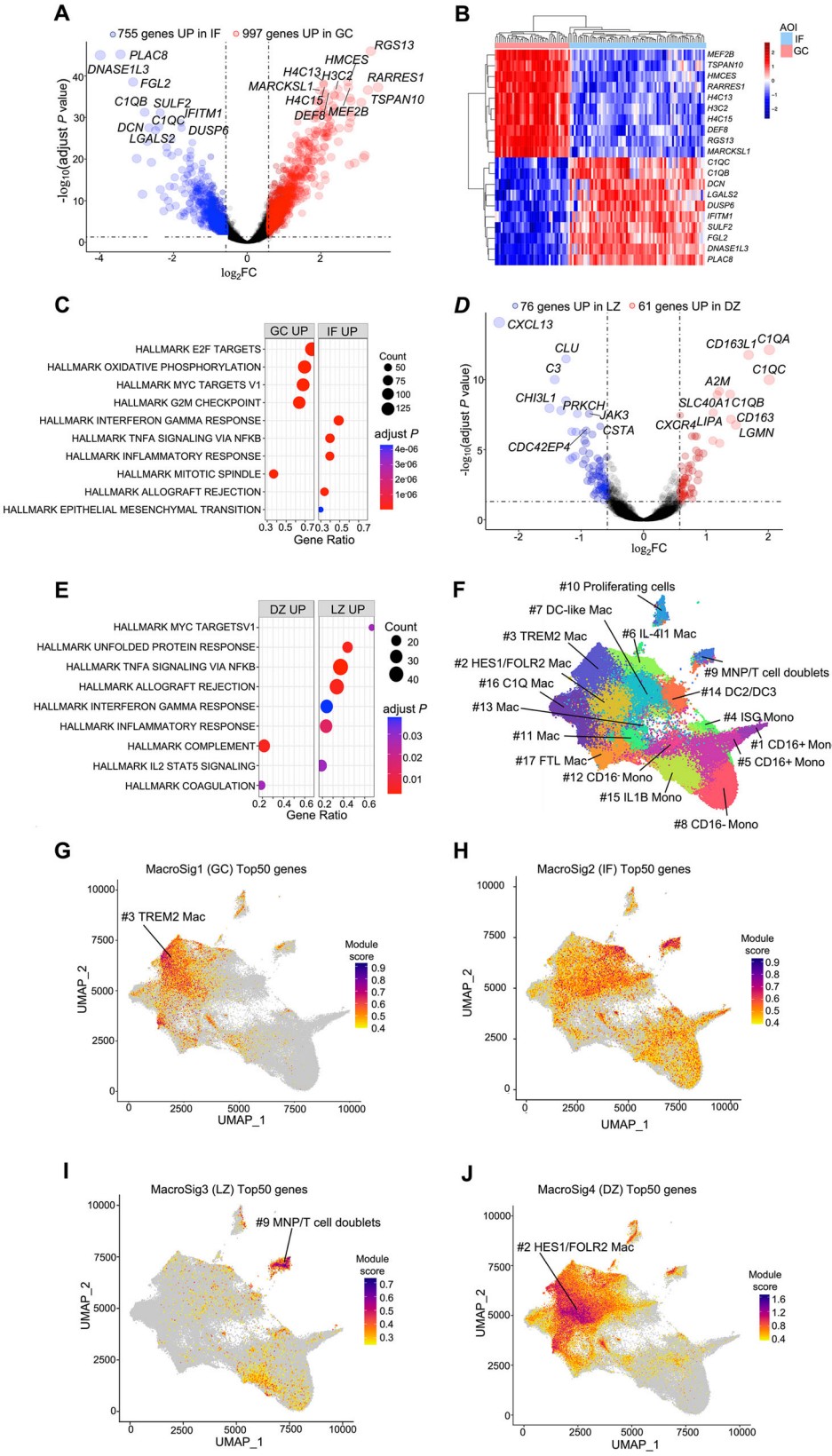

signature. Nonetheless, these results suggest that C1Q expressing macrophages populating the germinal center dark zone will be an important area of research of biology/ therapeutic relevance in DLBCL. Overall, our findings provide insights to the landscape of the DLBCL TME, and highlight the importance of dark zone biology in the prognosis of DLBCL.

## Discussion

Using DSP, through the application of a mask for CD68 (a broad marker for cells of macrophage-monocyte lineage)[43,44], we defined macrophage transcriptomic signatures in reactive and malignant lymphoid tissue (termed MacroSig1-6; corresponding to biological/ clinical categories listed in Table 1) and described their associations

**Fig. 2 | Unique gene expression patterns differentiate macrophages in distinct spatial locations within reactive lymphoid tissues. A** Volcano plot showing the DEGs of macrophages between the GC and IF based on adjusted $P < 0.05$ and |log$_2$FC| ≥ 0.58. $P$ values were determined by two tailed moderated $t$ test (BH corrected). **B** Top 20 macrophage DEGs (10 DEGs upregulated in GC and 10 DEGs upregulated in IF) are displayed based on adjusted $P$ value in the heatmap. **C** Pathway enrichment analysis was performed on all DEGs between GC and IF. $P$ value calculated by two tailed Fisher exact test (BH corrected). The top 10 pathways, based on BH adjusted $P$ value, are shown. **D** The volcano plot showed the

macrophage DEGs between LZ and DZ based on adjusted $P < 0.05$ and |log$_2$FC| ≥ 0.58. $P$ values were determined by two tailed moderated $t$ test (BH corrected). **E** Pathway enrichment analysis was performed on all macrophage DEGs between LZ and DZ. $P$ values were calculated by two tailed Fisher exact test (BH corrected). **F** MoMac-VERSE annotated 17 TAM subclusters using a compilation of 41 scRNA-seq datasets from 13 healthy and cancer tissues (Figure created via [https://macroverse.gustaveroussy.fr/2021_MoMac_VERSE/]). **G–J** Top50 genes of each MacroSig1-4 were projected respectively onto MoMac-VERSE. Germinal center, GC; interfollicular, IF; fold change, FC; macrophage signatures, MacroSigs.

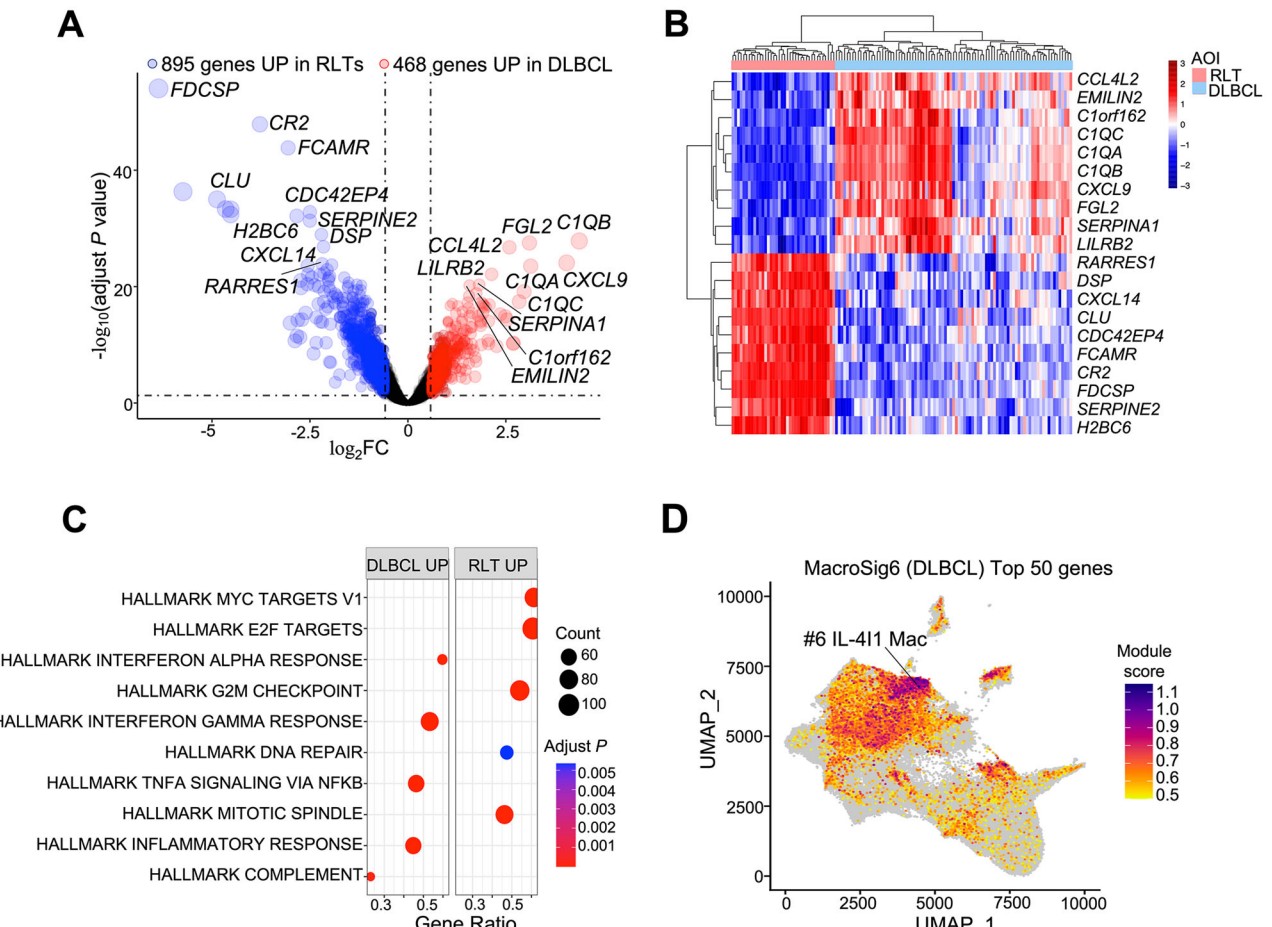

**Fig. 3 | Distinct transcriptomic profiles of macrophages between reactive and malignant lymphoid tissue. A** Volcano plot showing the macrophage DEGs between RLTs and DLBCL based on adjusted $P < 0.05$ and |log$_2$FC| ≥ 0.58. $P$ values were determined by two tailed moderated $t$ test (BH corrected). **B** Top DEGs between RLTs and DLBCL are displayed in the heatmap. **C** Pathway enrichment

analysis was performed on all macrophage DEGs between GC and DLBCL. $P$ values were calculated by two tailed Fisher exact test (BH corrected). The top 10 pathways, based on adjusted $P$ are shown. **D** Top50 genes of MacroSig6 (DLBCL) were projected onto MoMac-VERSE.

with known features of macrophage/DLBCL biology and clinical outcome (Fig. 7F). Macrophages are highly plastic, acquiring diverse phenotypes and functions that enable their role in a variety of physiological or pathological settings[45]. We demonstrate the spatial transcriptional diversity of macrophages within different regions of RLTs and between macrophages populating RLTs and DLBCL. While certain MacroSigs (GC, LZ, DZ, DLBCL) overlap with distinct and previously defined macrophage groups based on scRNA-seq[33], indicating that macrophage subpopulations in different spatial niches do have specific roles or functions within their respective regions, it is of interest that certain others do not (interfollicular [IF], RLT). As the reference scRNAseq datasets are not spatially resolved, it will be of interest to evaluate these MacroSigs in the future in spatially resolved reference datasets of macrophages. Distinguishing these relationships and

further orthogonal validation of these findings at single cell resolution would aid in the understanding of macrophage biology and the development of macrophage targeted therapies.

A key finding from our study is the association of certain MacroSigs with established clinically-relevant DLBCL subclassifications[5,46–48]. GEP identifies two prognostically distinct clusters of DLBCL based on COO, but there remains a consistent cluster of unclassified cases. We report a strong correlation of the signature of IF-macrophages (MacroSig2) with this unclassified cluster, along with clear correlations of MacroSig1 (GC) to the germinal center B-cell like (GCB) DLBCL cluster and MacroSig6 (DLBCL) to the ABC cluster. While the association of the germinal center (GC) MacroSig to the GCB B-cell COO signature may be explained by a high degree of similarity in cell-cycle/proliferative genes between these, the associations of the IF and DLBCL

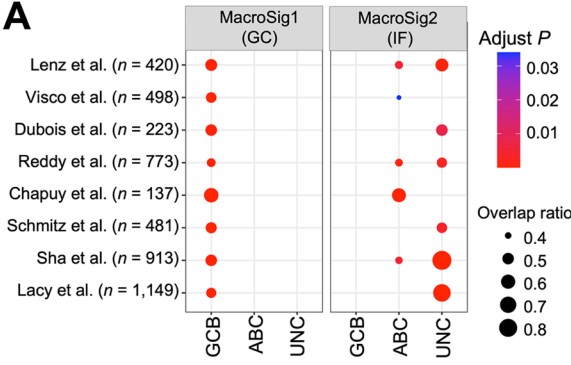

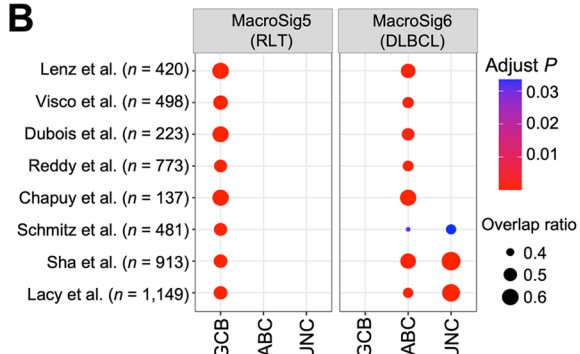

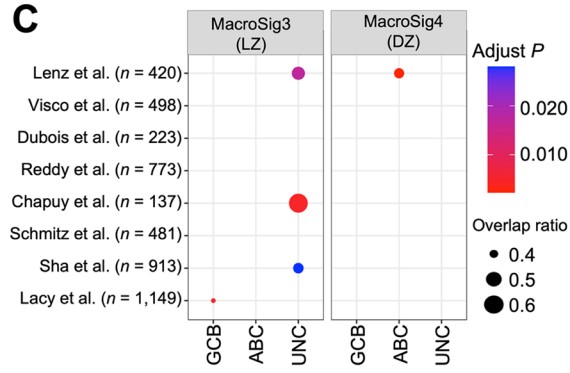

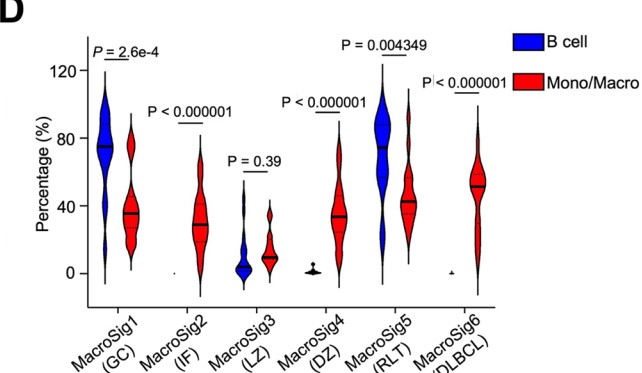

**Fig. 4 | Spatially-derived MacroSigs associate with COO DLBCL subclassifications.** The associations of MacroSigs with clinical categories (i.e., COO, genetic subtypes) were evaluated through the Fisher exact test. The overlap ratio refers to the number of patients classified as both a certain MacroSig and COO category, divided by the total number of patients classified in that particular COO category. **A** MacroSig1 (GC) and MacroSig2 (IF) were enriched in DLBCL COO classifications in bulk RNA gene expression profiles of DLBCL patients across eight publicly available transcriptomic datasets ($n$ = 4594, 8 datasets). **B** MacroSig5 (RLT) and MacroSig6 (DLBCL) were enriched in DLBCL COO classifications in the above-mentioned eight datasets. **C** MacroSig3 (LZ) and MacroSig4 (DZ) were not distinctly enriched in any

COO category in the above-mentioned eight datasets. **D** All genes of each MacroSig, through their respective module scores, were projected onto the Monocyte/Macrophage and B cell subsets of DLBCL scRNA-seq datasets (Ye et al; $n$ = 17). The violin plot depicts, for each patient, the percentage of B cells and macrophages expressing a given MacroSig (module score > 0.1) The median and quartile bands are depicted. $P$ values were calculated by a paired $t$ test (see also Supplementary Figs. 5 and 6). Source data are provided as a Source Data file. Germinal center B-cell like, GCB; activated B-cell like, ABC; unclassified, UNC; monocyte/macrophage, Mono/Mac.

MacroSigs to COO subtypes are particularly striking and supported by a negligible expression of genes from these MacroSigs in malignant B-cells. These results suggest that DLBCL gene expression signatures may be entwined with the nature of underlying macrophage infiltrates, and may indeed co-evolve during tumorigenesis. The role of IF-like macrophages in potentially determining the overall gene-expression status of the unclassified COO subcluster, and its links to TNFR signaling and inhibitory immune checkpoint overexpression[49] will be an interesting avenue of research.

We also identify MacroSigs that are prognostic for poor clinical outcomes in multiple independent DLBCL datasets. Though it was not the main objective of this study to define new predictors of DLBCL outcome to be used in routine practice, our results lend credence to the clinical significance of our signatures. For example, both MacroSig4 (DZ) and MacroSig6 (DLBCL) were negatively prognostic, and were associated with higher expression of M2-related markers such as *CD209*, *CSF1R*, *IL10*, and TGFβ-associated genes (*TGFBI*, *TGFB1*). Additionally, it is now recognized that macrophage phagocytosis checkpoint molecules are associated with resistance to rituximab in DLBCL[11]. We also found that macrophage checkpoints such as *SIRPα*, *LRLRB1*, *SIGLEC10*, and *PDCD1* are enriched in patients categorized to have MacroSig 6 (DLBCL) in comparison to MacroSig 5 (RLT) in eight DLBCL publicly available datasets (Supplementary Fig. 8). Future mechanistic work elucidating their underlying relationship could

contribute to the development of potential therapeutic strategies to overcome rituximab resistance.

Molecular high-grade (MHG)[50]/double-hit gene expression signature (DHITsig)[51] is another salient molecular subgroup of interest in DLBCL. This subgroup carries a uniformly poor prognosis and is thought to be reflective of GC-dark zone (DZ) biology[52,53], and was renamed the DZ signature[52]. It is intriguing that the strongest prognostic signature in our study was also of DZ-cells, but that of macrophages and not B-cells. Importantly, we found that genes comprising MacroSig4 (DZ), collected from CD68 + DZ macrophages, are distinct from published B-cell DZ signatures and from those collected in our own experiments from CD20 + DZ B-cells. Notably, both MacroSig4 (DZ) and B-cell DZ signatures were highly prognostic in multiple DLBCL datasets, with higher effect sizes and more consistent statistical significance for the MacroSig4 (DZ). None of genes constituting MacroSig4 (DZ) were noted to be highly expressed within CD20+ cells in DLBCL in our dataset, indicating that the signature is likely to confer its prognostic significance because of macrophage infiltration and not aberrant expression of these genes in tumor cells. Pathway enrichment analysis indicated that the cell proliferation/cycle-associated pathways were activated in DZ macrophages. A recent transcriptomic analysis of the circulating monocyte-derived macrophages (MDMs) identified a proliferative macrophage subcluster which influence MDMs-mediated inflammation and regeneration[54]. It is tempting to speculate a similar

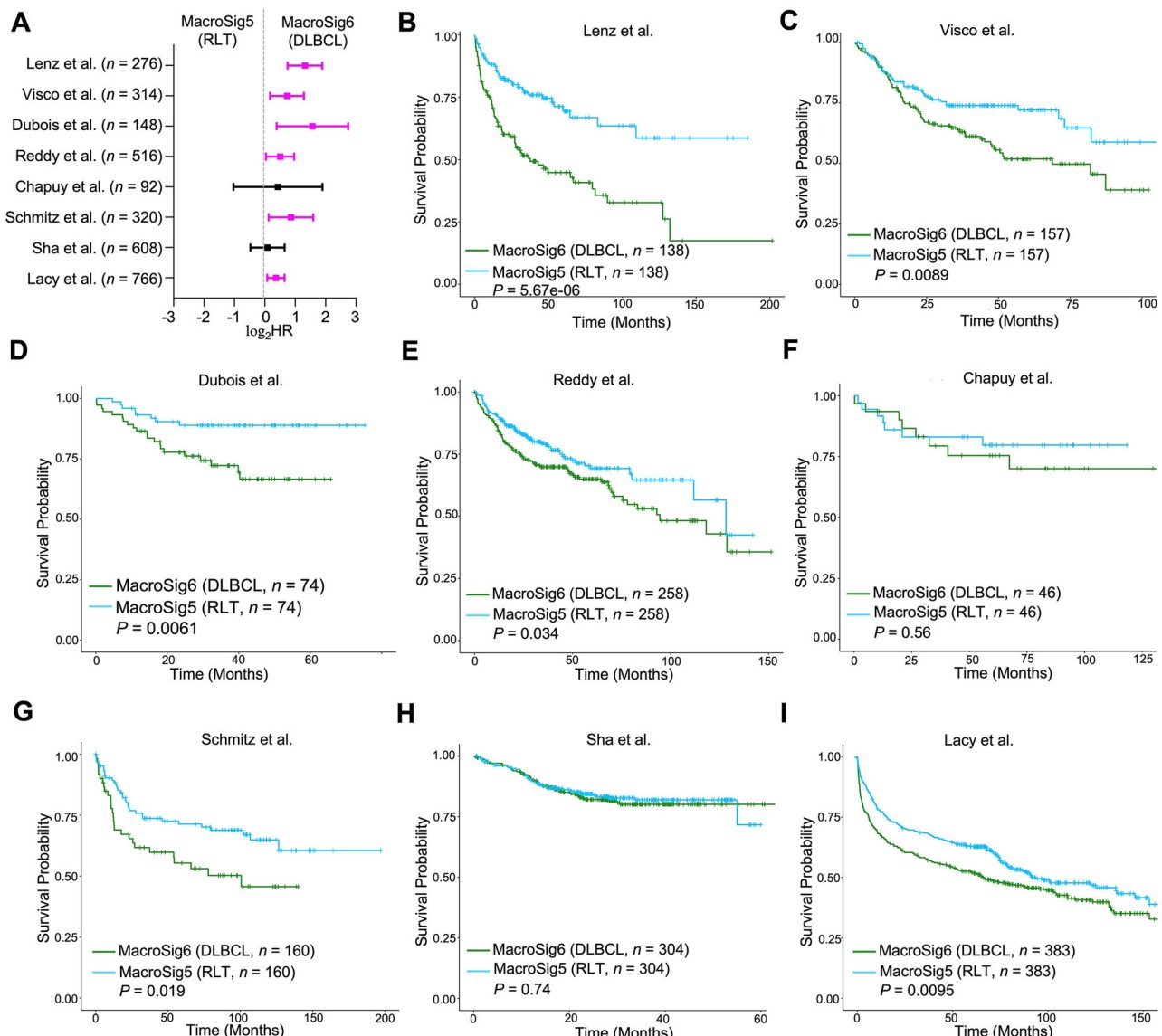

**Fig. 5 | Spatially-derived MacroSig5/6 (RLT/DLBCL) stratify for patient survival in DLBCL datasets. A** Forest plot depicting the univariate Cox proportional hazards model analysis, comparing MacroSig5 (RLT) and MacroSig6 (DLBCL) (represented as tertile groups, as described in Methods: Survival analysis). Analysis applied to bulk RNA gene expression profiles of DLBCL patients across eight publicly available transcriptomic datasets (n = 4594, 8 datasets). Data are presented as the 95% confidence interval of the hazard ratio (plotted in log-scale). Source data are provided as a Source Data file. **B–I** Kaplan–Meier analyses showed that patients with high expression of MacroSig6 (DLBCL) and low expression of MacroSig5 (RLT) were associated with poor OS across six distinct DLBCL datasets. P values generated by log-rank test. Overall survival, OS (see Methods: Survival analysis).

biology exists in these proliferative DZ macrophages, the molecular (proteomic/secreted) characteristics of which require further investigation.

A potential limitation of DSP is its reliance on average values across a group of cells defined by a cellular "mask", and its resolution which likely extends beyond the size of a single cell. There could exist heterogeneity within each mask, even within spatially defined regions, and this can hopefully be clarified through future advances in single-cell resolved spatial transcriptomic methods. Furthermore, the active scavenging function of macrophages may contribute to non-macrophage transcript contamination[55], which will be a general limitation of any spatial transcriptomic study of macrophages. Nonetheless, we have demonstrated that there is minimal overlap of gene sets between the CD68+ and CD20+ masks profiled regions, suggesting that neither phagocytosis nor the technical spill-over of transcripts between the CD68 and CD20 photocleaved DSP regions are likely to significantly contribute to the observed phenotypes associated with

different macrophage spatial profiles. Moreover, we also generated signatures by filtering transcripts potentially linked to close interactions between macrophages and T cells, to evaluate and exclude the contribution of transcripts related to the interaction with T lymphocytes, obtaining analogous results in terms of prognostic ability (Supplementary Fig. 9 and Supplementary Table. 4). Nevertheless, further refinements to spatial transcriptomics at single-cell resolution will provide more detailed dissection of intrinsic vs scavenged transcripts, and also allow direct evaluation of spatial interactions between different cell types, contributing to a deeper understanding of the actual relationships between macrophage subtypes and other TME components.

In summary, through the use of DSP, we present a spatially resolved transcriptomic characterization of macrophages in RLTs and DLBCL, showing diverse characteristics of macrophage landscapes in different spatial localizations. Spatially-derived MacroSigs of lymphoid tissue can complement existing genetic and molecular DLBCL

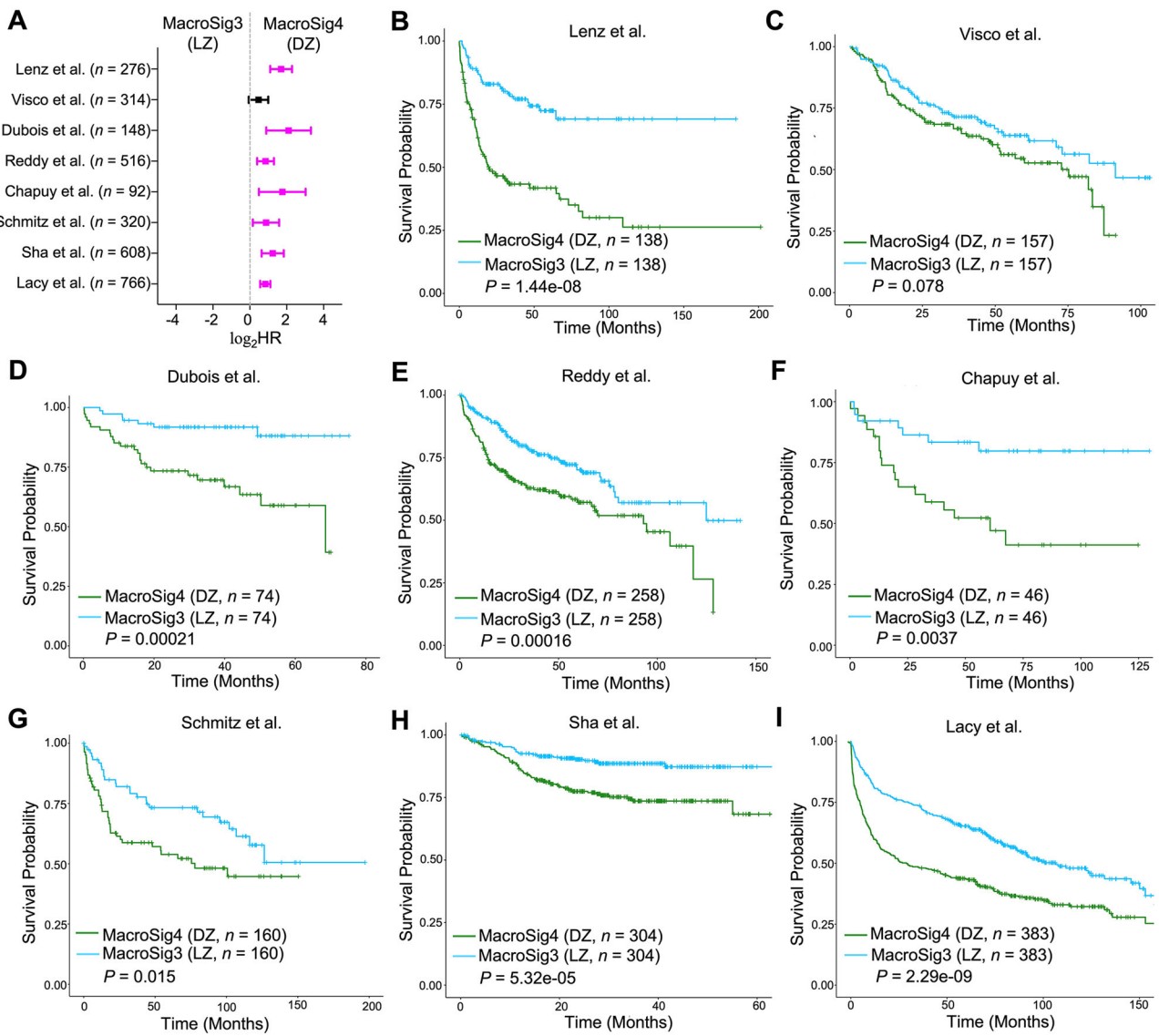

**Fig. 6 | Spatially-derived MacroSig3/4 (LZ/DZ) stratify for patient survival in DLBCL datasets. A** Forest plot depicting the univariate Cox proportional hazards model analysis, comparing MacroSig3 (LZ) and MacroSig4 (DZ) (represented as tertile groups, as described in Methods: Survival analysis). Analysis was applied to bulk RNA gene expression profiles of DLBCL patients across eight publicly available transcriptomic datasets (*n* = 4594, 8 datasets). Data are presented as the 95% confidence interval of the hazard ratio (plotted in log-scale). Source data are provided as a Source Data file. **B–I** Kaplan–Meier analyses showed that patients with high expression of MacroSig4 (DZ) and low expression of MacroSig3 (LZ) were associated with poor OS in DLBCL patients across seven distinct DLBCL datasets. *P* value generated by log-rank test.

subclassifications, contributing to our understanding of the DLBCL TME and the ever-evolving classification of this heterogenous disease, and are also prognostic across numerous distinct DLBCL datasets. These data provide a framework to further evaluate the biological and clinical relevance of macrophage subtypes in lymphoid biology and disease.

## Methods

### DSP study population

Our research complies with all relevant ethical regulations. All biopsy samples were pre-treatment samples and obtained from the Department of Pathology, National University Hospital, with IRB approved waiver of consent in accordance with the ethical guidelines of the National Healthcare Group domain specific review board (NHG DSRB) approved protocol 2015/00176. This waiver of consent applies to all samples obtained between 1st January 1990 and 30th April 2020 on the basis that there is no longer patient contact (patient is deceased or lost

to follow-up) and that this study poses minimal risk to the patient. The overall DSP study population was made up of two patient groups. Group 1 comprised of two tissue microarrays (TMAs) of de novo DLBCL samples derived from pre-treatment biopsies of 64 patients between 2010 and 2017 subsequently treated with 6x R-CHOP with a follow-up time more than three years (for relapsed patients, the follow-up time is at least 1 year), at the National University Hospital in Singapore (Supplementary Fig. 10). 23 patients had duplicate cores between both TMAs, meaning a total of 87 biopsies were profiled. Patient details and characteristics from the aforementioned cohorts are summarized in Supplementary Table 5. Group 2 (non-malignant RLT samples) comprised of a TMA containing 12 tonsil samples and 11 tonsil whole-slide samples, obtained from patients with tonsillectomies at the National University Hospital for non-cancer indications. Also included into Group 2 was a whole-slide tonsil section retrieved from the archives of the Tumor Immunology Laboratory of the University of Palermo and approved by the University of Palermo

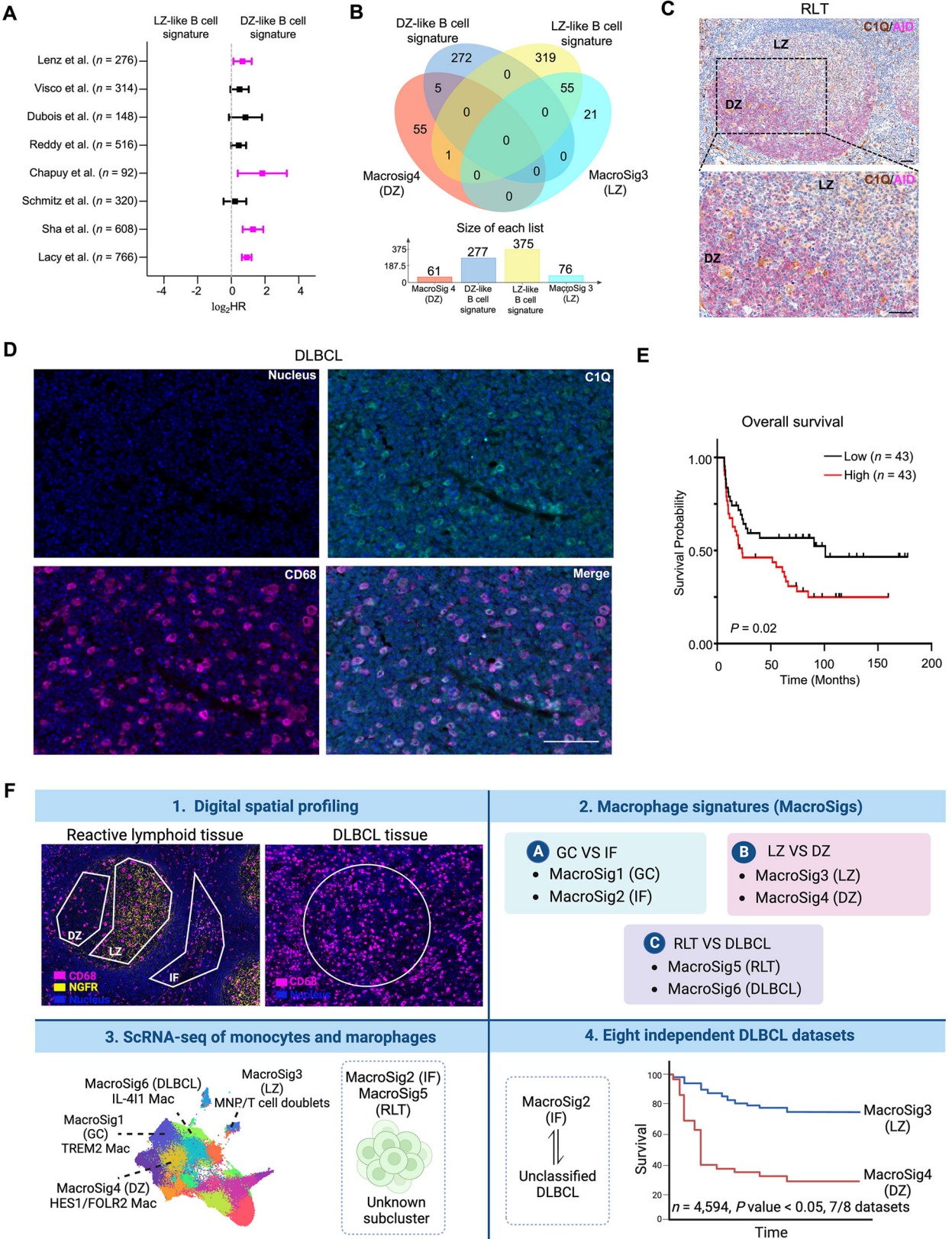

Institutional Review Board 09/2018. All tissues were formalin-fixed and paraffin-embedded at the time of resection and stored as paraffin blocks. An additional DLBCL TMA from the Chi-Mei Medical Center, Taiwan (CMMC cohort; $n = 86$) with OS more than 6 months, was used as a validation cohort for quantitative immunofluorescence analyses. Usage of tissues from all providing institutions is incorporated into the framework of an NUS IRB approved translational study (H-19-055E).

**GeoMx® DSP WTA assay**
Spatial transcriptomics of RLTs and DLBCL tissues was conducted using the GeoMx® WTA kit (GMX-RNA-NGSHuWTA-4, NanoString,

**Fig. 7 | Additional evaluation of the Dark Zone MacroSig hallmark C1Q in DLBCL. A** Forest plot depicting the univariate Cox proportional hazards model analysis, comparing B cell-based LZ and DZ signatures (represented as tertile groups, as described in Methods: Survival analysis). Analysis applied to bulk RNA gene expression profiles of DLBCL patients across eight publicly available transcriptomic datasets ($n = 4594$, 8 datasets). Data are presented as the 95% confidence interval of the hazard ratio (plotted in log-scale). Source data are provided as a Source Data file. **B** Venn diagram displaying the overlapping genes of LZ-, DZ-like B-cell signatures, and MacroSig3-4 (LZ and DZ). **C** Immunochemistry staining of RLTs was shown ($n = 3$). Activation-induced cytidine deaminase (AID) in magenta

was used for illuminating the LZ and DZ. C1Q in brown stained macrophages. Scale bar: 100 μm. Source data are provided as a Source Data file. **D** Immunofluorescence stained CD68 + C1Q+ cells in DLBCL tissues ($n = 86$). Representative images are shown. Scale bar: 100 μm. Source data are provided as a Source Data file. **E** Kaplan–Meier analyses showed that patients with highly infiltrating levels of CD68 + C1Q+ cells were associated with poor OS in DLBCL patients in CMMC cohort. *P* value generated by log-rank test. **F** Graphical abstract summarizing the derivation of the spatial derived MacroSigs and describing their associations with known features of macrophage/ DLBCL biology and clinical outcome (created with BioRender.com).

### Table 1 | Six distinct MacroSigs based on biological/clinical characteristics

| MacroSigs | Abbreviations | Associated biological/clinical feature |
|---|---|---|
| MacroSig1 | GC-MacroSig | Macrophages from GC of RLTs |
| MacroSig2 | IF-MacroSig | Macrophages from IF of RLTs |
| MacroSig3 | LZ-MacroSig | Macrophages from LZ of RLTs |
| MacroSig4 | DZ-MacroSig | Macrophages from DZ of RLTs |
| MacroSig5 | RLT-MacroSig | Macrophages from RLTs |
| MacroSig6 | DLBCL-MacroSig | Macrophages from DLBCL patients |

*MacroSigs* macrophage signatures, *GC* germinal center, *IF* interfollicular, *LZ* light zone, *DZ* dark zone, *RLTs* reactive lymphoid tissues, *DLBCL* diffuse large B-cell lymphoma.

Seattle, Washington, USA), according to manufacturer's instructions (Fig. 1A).

**GeoMx® DSP sample preparation on Bond Max.** The formalin-fixed paraffin-embedded sections were freshly cut (5 μm thick) and placed on Bond plus slides (S21.2113.A, Leica Biosystems, Germany). The slides were baked at 60 °C for 1 h and loaded, with covertiles, into the slide tray on Bond Max Fully Automated IHC and ISH Staining System for deparaffinization, rehydration, antigen retrieval (ER2 solution [AR9640, Leica Biosystems] at 100 °C for 20 min), RNA digestion (Proteinase K 1 μg/ml for 15 min) and post-fixation (10% neutral buffered formalin [NBF, HT501128, Sigma-Aldrich, St. Louis, Missouri, USA] for 5 min, NBF stop buffer for 5 min twice). The NBF stop buffer was prepared using Tris base (H5133, Promega, Madison, Wisconsin, USA) and Glycine (15527013, Thermo Fisher Scientific, Waltham, Massachusetts, USA) in DEPC-treated water. Upon run completion, the covertiles were removed and the slides were soaked in PBS for subsequent hybridization.

**In-situ hybridization.** An overnight in-situ hybridization was performed with GeoMx® Human NGS WTA (GMX-RNA-NGSHuWTA-4, NanoString) that contained probes for 18,000+ protein-coding genes. After which, the slides were washed twice with equal parts of 4X SSC (15557044, Thermo Fisher Scientific) and 100% formamide (AM9342, Sigma-Aldrich) at 37 °C for 25 min to remove off-target probes.

**GeoMx® DSP sample collection.** The slides were incubated with blocking Buffer W (200 μL/slide, GMX-PREP-RNAFFPE-12, NanoString) for 30 min in the humidity chamber at room temperature after hybridization. For Group 1, slides of DLBCL TMA were stained with macrophage marker CD68 (sc-20060 AF594, Santa Cruz biotechnology, Texas, USA), T-cell marker CD3 (A0452, Dako, California, USA), B-cell marker CD20 (NBP2-47840 AF647, Novus Biologicals, Colorado, USA) and the nuclear stain SYTO 13. For Group 2 (Fig. 1B), the slide of RLT TMA was visualized with CD68, the follicular dendritic cell marker nerve growth factor receptor (Ab52987, NGFR, Abcam, Cambridge, UK) and SYTO 13. Individual RLT sections of Group 2 (Supplementary Fig. 1C) were stained with similar markers as Group 1. Additionally, corresponding serial sections were stained with NGFR to identify light zone (LZ) and dark zone (DZ) regions. After immunofluorescent staining, the slides were visualized using the GeoMx® DSP instrument

(software: 2.4.0.421) to select regions of interest (ROIs). To acquire representative regions, ROI selection was performed by an expert pathologist. Based on the intensity of respective morphology marker's fluorescent staining (CD68, CD3, and CD20), we adjusted the thresholds of each channel until the desired masks are generated to fit the corresponding cell types precisely. Then each ROI was segmented into corresponding areas of interest (AOIs) - CD68+ regions, CD3+ regions, and CD20+ regions based on the respective masks. Subsequently, each AOI- defined by the cellular masks within an ROI (e.g., CD68)- was exposed to UV light and photocleaved oligos were aspirated from the solution into the wells of a collection plate (Fig. 1A) for downstream sequencing and data processing. We therefore obtain cell-type specific transcriptome data for all cells marked by the mask, across a range of AOI sizes. The gene counts obtained from each AOI are the aggregate from all cells of a given cell type within an ROI (normalized as described in Methods: DSP data processing and harmonization). The size and cell numbers of each AOI are supplied in Supplementary Data 4.

**Library preparation and sequencing.** Collected photocleaved oligos were PCR amplified with the corresponding GeoMx® Seq Code Primer Plate and Master Mix (GMX-NGS-SEQ, NanoString). PCR products were pooled and cleaned with AMPure XP beads (A63880, Beckman Coulter, Brea, California, USA) twice to obtain the libraries. The quality and concentration of libraries were assessed using a high sensitivity DNA Kit (5067-4626, Agilent, Santa Clara, California, USA) and Bioanalyzer. Subsequently, libraries were sequenced on an illumina sequencing platform (HiSeq 3000 or NovaSeq 6000) with standard workflow specifications (dual-indexing and paired-end reads [$2 \times 27$ bp]).

### DSP data processing and harmonization

Raw reads were trimmed, stitched, aligned, and deduplicated to generate digital counts data for unique target genes. AOIs with fewer than 10,000 raw reads or sequencing saturation <50% were filtered out of the analysis. AOIs with less than 5% of all target genes (18,000 +) and target genes that did not achieve the limit of quantitation were removed. The data was then processed with the Q3 normalization method for all the remaining targets according to NanoString guidelines. Q3 normalization divides the counts in one AOI by the 3rd quartile value for that AOI, then subsequently multiplies that value by the geometric mean of the 3rd quartile values of all AOIs. Q3 normalization rescales the gene expression data such that all AOIs have similar

gene expression ranges. It reduces variance from AOI size, AOI cellularity, and other technical factors. Underlying the Q3 method is an assumption that, despite biological variation, AOIs should have similar gene expression count distributions. Thus, Q3 is only appropriate when a probe panel is large and diverse, such as the one used here, that targets the full transcriptome.

As the spatial profiling was conducted as two individual experiments, the sequencing data was harmonized using the removeBatchEffect function of the limma (3.56.2) package based on the respective masks (CD68, CD20, and CD3)[56]. Principal component analysis (PCA) analysis was conducted using the FactoMinR (2.9) package, and outlier AOI were removed according to the PCA projections. In addition, the Kolmogorov-Smirnov test was applied to further assess the overall distribution of the data obtained from each mask: Using a procured set of genes representing macrophages (*CD68, CD163, FCGR1A*, and *CSF1R*), T cells (*CD3D, CD3E, UBASH3A, CD2*, and *TRBC2*), and B cells (*MS4A1, CD79A, CD79B, CD19*, and *PAX5*) the cumulative expression for each signature was evaluated and compared within each mask (Fig. 1D, E). Following which, formal analyses were conducted as detailed in subsequent sections. The overall data analysis workflow is provided in Supplementary Fig. 11. All statistical analyses were performed using R statistical software (v 4.3.0) (http://www.R-project.org).

## Clustering analysis

The Euclidean distance metric across AOIs was considered for the sample hierarchical clustering analysis, and the ward.D2 aggregation method was used to build the heatmap dendrogram within the R package pheatmap (1.0.12).

## Differential expression analyses (DEAs) and derivation of MacroSigs

DEAs were conducted using moderated *t* test from the limma package[56]. The presence of DLBCL samples that belong to the same patient was considered using the duplicateCorrelation option of the limma package. Upregulated/downregulated genes were selected by applying the Benjamini-Hochberg correction on the *P* values (BH adjusted $P < 0.05$) and considering the $|\log_2 FC| > 0.58$. Based on both criteria, MacroSigs1-6 were derived from the DEAs between GC/IF, LZ/DZ, and RLT/DLBCL To establish the representative genes of our MacroSigs and filter out contaminant genes derived from B-cells, we removed those DEGs whose percentile rank of average expression, as determined after Q3 normalization, was greater in the CD20 mask than in the CD68 mask. The resulting DEGs constituted the corresponding MacroSigs. This filtering process was not applied to the MacroSig5 (RLT) due to the absence of data from corresponding CD20 masks within whole-region germinal center ROIs.

## Pathway enrichment analyses

The pathway enrichment analyses were performed considering the Hallmark gene sets downloaded from Human Molecular Signature Database (MSigDB)[57]. The Fisher exact test *P* values were calculated using the phyper function of the R software and adjusted for multiple comparisons applying the BH correction. The dot-plots were generated using clusterProfiler (3.12), org.Hs.eg.db (3.18.0), and ggplot2 (2 3.4.4) R packages. Count refers to the number genes present in the overlap between the MacroSigs and the Hallmark gene sets. The gene ratios were obtained by dividing the count by the total number of genes in that respective Hallmark gene set.

## Single-cell sequencing data analysis

Seurat (2.3.0)[58] was used for the analysis of the single-cell RNA sequencing (scRNA-seq) datasets. All functions were run with default parameters, unless specified otherwise. Low quality cells (< 200 genes/cell and >10% mitochondrial genes) and genes that were present in fewer than 3 cells were excluded. Clusters were defined based on

annotations provided by original authors (refs. 33,37,59). Refinement and validation of annotation was conducted by projecting and evaluating a curated B cell, T cell and macrophage marker list (see Methods: DSP data processing and harmonization).

For characterization and feature expression analysis of the MacroSigs, each MacroSig was further refined. A module score (https://satijalab.org/seurat/reference/addmodulescore) using the top 50 statistically significant DEGs, ranked by their log fold change, was created for each MacroSig. This score was then projected onto the uniform manifold approximation and projection (UMAP) space named MoMacVERSE. Moreover, to determine that all genes of our MacroSigs belong to macrophages, all genes of each MacroSig, through their respective module scores, were projected onto the Monocyte/Macrophage and B cell subsets of DLBCL scRNA-seq datasets ($n = 17$)[37] and subsequently compared (Fig. 4D and Supplementary Figs. 5, 6).

## Survival analysis

To evaluate the predictive power of our MacroSigs from a clinical standpoint, we applied them to eight distinct DLBCL bulk gene expression cohorts[38–40,46,50,60–62]. The MacroSigs were tested in pairs: GC/IF, LZ/DZ, RLT/DLBCL. In line with the percentile strategy commonly used in the literature[63], we divided the patients into tertiles based on MacroSig scores calculated by the following formula:

$$\text{score} = \sum_{i=1}^{n} -\log_{10}(p_i) \cdot x_i \cdot I_i \tag{1}$$

$$I_i = \begin{cases} -1, & \text{if } \log_2 FC_i < 0 \\ +1, & \text{if } \log_2 FC_i > 0 \end{cases} \tag{2}$$

Where $p_i$ and $FC_i$ are the moderated *t*-test *P* value and *FC* of gene-*i* obtained from the DEAs, $x_i$ is the expression of gene-*i* from the DLBCL bulk RNA-seq data, and $n$ is the number of genes within the gene signature. The $\log_{10}(p_i)$ quantity was introduced to weigh the genes in relation to their significance level. Comparing high and low tertile groups increases the chance of detecting biomarker related differences between the two groups. Therefore, based on the score derived from a paired MacroSig, e.g., DZ [MacroSig4]/LZ[MacroSig3]: the aggregate of genes upregulated in the DZ (MacroSig4) and downregulated in the DZ (MacroSig3 [LZ]), each DLBCL cohort was divided into three groups based on tertile values: (1) patients having high expression of downregulated genes and low expression of upregulated genes [e.g., patients classed as MacroSig3]; (2) patients having an intermediate gene signature expression. (3) patients having high expression of upregulated genes and low expression of downregulated genes [e.g., patients classed as MacroSig4]. The extreme groups (i.e., group 1 [e.g., MacroSig3] and group 3 [e.g., MacroSig4]) have been compared in terms of OS, COO, genetic subtypes categories, and DEA of macrophage checkpoints.

For a comprehensive survival analysis, we applied both the Cox proportional hazards model and the Kapan-Meier method. Before fitting the Cox model and conducting the log-rank test, the cox.ph test was used to test the proportional hazard assumption. We applied the Cox model to MacroSigs as a discrete variable, comparing the top tertile group with the bottom tertile group, as described above, to determine the hazard ratio associated with each MacroSig (displayed as forest plots). The Kaplan-Meier method was used to estimate the survival functions among tertile groups, and the log-rank test was used to test the differences in the OS between groups. The survival (3.5–7) and survminer (0.4.9) R packages were used for the survival analysis estimations.

## Association analysis

The associations between patient groups and clinical categories (i.e., COO, genetic subtypes, microenvironment categories) were evaluated through the Fisher exact test. The $P$ values were calculated using the phyper function of the R software and adjusted for multiple comparisons applying the BH correction. The COO association analyses are presented as dot plots, where the overlap ratio refers to the number of patients classified as both a certain MacroSig and COO category, divided by the total number of patients classified in that particular COO category (Fig. 4A–C). The genetic subtypes association analysis is presented as an integrated bar graph, where the strength of association between MacroSigs and genetic subtypes is represented by an enrichment score calculated by: -log10 (adjusted Fisher $P$ value) (Supplementary Fig. 7A). The microenvironment categories association analysis is displayed in the dot plot (Supplementary Fig. 7B). Count refers to the number genes present in the overlap between the MacroSigs and DLBCL microenvironment categories generated by ref. 5. The overlap ratios were obtained by dividing the count by the total number of genes in that respective DLBCL microenvironment category.

## Software

Unless otherwise stated, graphs were constructed with GraphPad Prism (9.5.0; GraphPad Software, Massachusetts, USA).

## Reporting summary

Further information on research design is available in the Nature Portfolio Reporting Summary linked to this article.

## Data availability

The DSP RNA-seq raw data and processed data generated in this study have been deposited in GEO under accession code GSE232853. DLBCL expression array datasets of ref. 46 (GSE10846, $n = 420$), ref. 62 (GSE31312, $n = 498$), ref. 60 (GSE87371, $n = 223$), ref. 40 (GSE98588, $n = 137$), ref. 50 (GSE117556, $n = 913$), and ref. 39 (GSE181063, $n = 1149$) were obtained from Gene Expression Omnibus. DLBCL RNA-seq dataset, ref. 61 ($n = 773$), was obtained through The European Genome-phenome Archive (EGA) at the European Bioinformatics Institute, Study ID: EGAS00001002606. This is a restricted access dataset that can be obtained through a data access request application as instructed on the EGA portal (https://ega-archive.org/). Ref. 38 ($n = 481$) was obtained from the National Institutes of Health (NIH) database of Genotypes and Phenotypes (dbGaP), accession number: phs001444.v2.p1. This is a restricted access dataset that can be obtained through a data access request application as instructed on the dbGaP portal (https://www.ncbi.nlm.nih.gov/gap/). Single-cell RNA sequencing (scRNA-seq) datasets of tonsil were obtained from HCATonsilData[59] [https://bioconductor.org/packages/release/data/experiment/html/HCATonsilData.html]. The scRNA-seq dataset of DLBCL from ref. 37 ($n = 17$) is available at the CNGB Sequence Archive (CNSA) of the China National GeneBank DataBase (CNGBdb) under accession number CNP0001940. This is a restricted access dataset that can be obtained through a data access request application as instructed on the CNGBdb portal (https://db.cngb.org/). The integrative scRNAseq dataset for human monocytes and macrophages (MoMac-VERSE; from 41 scRNA-seq datasets comprising 13 healthy and pathological tissues), was obtained from ref. 33 [https://macroverse.gustaveroussy.fr/2021_MoMac_VERSE/]. The above-mentioned $n$ in this section refers to patient numbers. Source data are provided with this paper.

## Code availability

The codes used for our manuscript are available in Zenodo (https://doi.org/10.5281/zenodo.10511030)[64] or github (https://github.com/shrutisridhar99/Spatially-resolved-transcriptomics-of-macrophage-heterogeneity-and-prognostic-significance-in-DLBCL.git).

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

## Acknowledgements

M.L. was supported by the China Scholarship Council (202006940018). A.D.J. was supported by the Singapore Ministry of Health's National Medical Research Council Clinician Scientist Award (MOH-000715-00). Work in A.D.J.'s laboratory is funded by a core grant from the Cancer Science Institute of Singapore, National University of Singapore through the National Research Foundation Singapore and the Singapore Ministry of Education under its Research Centres of Excellence initiative. The

Genomic Variation in Diffuse Large B Cell Lymphomas study, from which the data presented in ref. 38 was derived, was supported by the Intramural Research Program of the National Cancer Institute, NIH, Department of Health and Human Services. CT was supported by the Italian Foundation for Cancer Research (AIRC) Investigator Grant IG ID.22145; 5×1000 Grant ID.22759, and the Italian Ministry of Education, University and Research (MIUR) Grant 2017K7FSYB. GB was supported by Italian Ministry of Education, University and Research (MIUR) through the "PON Research and Innovation 2014–2020".

## Author contributions

Conceptualization: ML, CT, ADJ Methodology (Experimental and analytical design): ML, CT, ADJ Project administration (sample provision/ethics approvals): RC, J, YB, SDM, LP Investigation (DSP, Quantitative mfIHC staining/ imaging): ML, YP, FS Data curation (Collection and curation of molecular and clinical data): RC, J, YB, JT, SDM, LP, EHLC, JL, YLC, MMH, KM, FG, XY, QP Formal analysis (Pathological review of cases): SSSH, SBN, STC, SSC Formal analysis (Multiplex analysis): ML, YP Formal analysis (Bioinformatics/statistical analysis): GB, ShS, KM Interpretation of findings: ML, CT, ADJ Writing – original draft: ML, CT, ADJ, SBN, NS, RXL, KGC Writing – review & editing: ML, CT, ADJ, SBN, NS, RXL, PJ Underlying data verification: ML, CT, ADJ.

## Competing interests

A.D.J. has received consultancy fees from DKSH/Beigene, Roche, Gilead, Turbine Ltd, AstraZeneca, Antengene, Janssen, MSD and IQVIA; and research funding from Janssen and AstraZeneca. The other co-authors have no conflicts of interest to declare.

## Additional information

[1]Cancer Science Institute of Singapore, National University of Singapore, Singapore, Singapore. [2]Department of Radiation Oncology, Chongqing University Cancer Hospital, Chongqing, PR China. [3]Department of Immunology, Tianjin Medical University Cancer Institute and Hospital, Tianjin, PR China. [4]Department of Economics, Business and Statistics, University of Palermo, Palermo, Italy. [5]Tumor Immunology Unit, Department of Sciences for Health Promotion and Mother-Child Care "G. D'Alessandro", University of Palermo, Palermo, Italy. [6]Singapore Immunology Network, Agency for Science, Technology and Research, Singapore, Singapore. [7]Institut National de la Santé Et de la Recherche Medicale (INSERM) U1015, Equipe Labellisée—Ligue Nationale contre le Cancer, Villejuif, France. [8]Université Paris-Saclay, Gustave Roussy, Villejuif, France. [9]Department of Pathology, Yong Loo Lin School of Medicine, National University of Singapore, Singapore, Singapore. [10]Department of Biomedical Informatics, Yong Loo Lin School of Medicine, National University of Singapore, Singapore, Singapore. [11]Department of Haematology-Oncology, National University Health System, Singapore, Singapore. [12]NUS Centre for Cancer Research, Yong Loo Lin School of Medicine, National University of Singapore, Singapore, Singapore. [13]Department of Pathology, Chi-Mei Medical Center, Tainan City, Taiwan, ROC. [14]Lee Kong Chian School of Medicine, Nanyang Technological University Singapore, Singapore, Singapore. [15]Kindstar Global Precision Medicine Institute, Wuhan, PR China. [16]Division of Immunology, Department of Medical Biochemistry and Biophysics, Karolinska Institutet, Stockholm, Sweden. [17]Histopathology Unit, Institute of Molecular Oncology Foundation (IFOM) ETS - The AIRC Institute of Molecular Oncology, Milan, Italy. [18]Department of Medicine, Yong Loo Lin School of Medicine, National University of Singapore, Singapore, Singapore. ✉e-mail: claudio.tripodo@unipa.it; csiadj@nus.edu.sg

