## [Peer Review File · Nature Communications]

REVIEWER COMMENTS

Reviewer #1 (Remarks to the Author): Expert in DLBCL genomics and clinical research

In this work, Liu and colleagues use spatial transcriptomics to explore tumor microenvironment in DLBCL, tumor-associated macrophages in particular. The authors conducted a comparative mapping of macrophages gene expression patterns within FFPE tissues of reactive lymphoid tissues (RLT; n=17) and diffuse large B-cell lymphoma (DLBCL; n=47). The digital spatial profiling with whole transcriptome performed on CD68+ cells in different conditions provided differentially expressed genes (DEG) that were projected to a public single cell RNA-seq macrophage atlas, leading to the definition of 8 spatially-derived macrophage gene expression signatures (named MacroSig 1 to 8). MacroSig#1 corresponds to germinal center (GC), #2 to interfollicular (IF) region, #3 to light zone of GC, #4 to dark zone of GC, #5 to reactive lymphoid tissue (all GC areas), #6 to DLBCL, #7 to non-relapsed DLBCL and #8 to relapsed DLBCLs. Some of them are associated with survival when applied on publicly available DLBCL datasets.

This is an interesting topic, explored with innovative tools. However, in its current form, some aspects of this article are not easily accessible to the readership. The methods used to define the different macrophages gene expression signatures are not fully convincing and should be better explained. Only a few insights into biological features are provided and the underlying DLBCL tumor microenvironment is not extensively explored. I am not sure that the prognostic markers provided in this work will emerge from the plethora of already available DLBCL predictive markers.

Major issues:

1.a. the results are supported by small sample sizes, and derived statistics for determining DEG between the different groups seems insufficiently stringent for manipulating such a high amount of data. The authors used moderated t-test, Fisher exact test, with a significant p-value set at 0.5. To be selected as a differentially expressed between two groups, a gene must meet at least one of the 2 following criteria : (i) adjusted p-value < .5 after Benjamini-Hochberg correction, (ii) an absolute \log_2 (fold change) > 0.58. So the fold change is sufficient for selection without passing a statistical test.

b. I am a little skeptical about classifying gene expression levels into 3 groups (high, intermediate and low) and compare survival only between the 2 extreme groups for subsequent survival analyses. I did not find the classification threshold for these three groups, even in the supplementary methods. Why removing the intermediate group from the analysis?

2.a. Little is told about sample selection: how was the sample selection conducted (supplemental methods only mention "random selection") ? Among DLBCLs, did the authors include de novo DLBCLs only or are there also transformed DLBCLs?

b. How did the author construct the non-relapsed DLBCL group with a ~ 6-year median follow-up? A subset of DLBCL relapses emerge later. What is the method used to assess relapse? Did they all undergo FDG-PET/CT? Were DLBCL patients sub-classified into transcriptomic-based ABC or GCB categories? Or in the genetic-based DLBCL subsets more recently identified?

3.a. Regarding tissue sample selection, did the authors use only lymph node biopsies for DLBCLs? Since the authors worked on bulk tissues, it is of major importance to accurately determine the relative

proportion of tumor cells and microenvironment cells. At least, was the tumor purity assessed (large cells component)? Did the authors perform a pathology review to confirm the diagnosis ?

b. It is stated that "macrophages are abundant cells in the microenvironment of DLBCL". Was an objective test performed to evaluate it properly for each sample (at least to ensure that this proportion is equivalent in all tested samples)?

c. Figure 1D: it seems that CD68 is not fully macrophage-specific, as there is a CD68-originating component in the T-cells and in the B-cells signature, suggesting that a portion of this CD68 signature is wrongly attributed to these cell types.

Moreover, figure 2C: E2F (a B-cell transcription factor) targets were the most enriched pathway in GC macrophages.

Do the authors consider that CD68 staining is specific enough to accurately and specifically select macrophages within tissues ?

d. Last, the lymph node architecture is commonly highly disrupted in DLBCL, how do the authors delineate CD68+ areas in this context, with a widely dominant tumor cell content ?

4.a. In devising their macrophage gene-expression signatures, the authors used comparative transcriptomics. The design is not clear, and some groups are overlapping : first the authors compared GC versus interfollicular area (MacroSig 1 and 2), then the light zone versus dark zone (2 components of the GC ; MacroSig 3 and 4), then RLT (ie light zone, dark zone, GC and interfollicular area) versus DLBCL (MacroSig 5 and 6) and ultimately non-relapsed versus relapsed DLBCL (MacroSig 7 and 8). The authors should better explain the design used to define these signatures, and how the associated statistical methods support these results.

b. Results, first paragraph : in RLTs' GC, light zone- and dark zone-originating macrophages present with no statistically significant differentially expressed genes (DEG). Then how could the authors derive MacroSigs from this? Later in the results section, and also in the summary: DZ MacroSig (#4) is associated with shorter survival while LZ MacroSig (#3) is not. How do you explain this? The same applies to MacroSig#7 and #8, as no DEG passed the significance threshold after p-value correction.

c. When projected on publicly available datasets, MacroSig#6 (derived from RLT (GC area) versus DLBCL comparison) was enriched in ABC subtype DLBCLs. It seems predictable, as genes highly expressed in RLT's GC were removed from MacroSig#6.

d. MOMac-VERSE : this tool was used to project MacroSigs on a gene expression-derived map of 17 annotated monocytes/macrophages subtypes. But most of the times, this does not lead to a clear labelling, except for MacroSig#7, and MacroSig#8, which seem to cluster within #2 HES1/FOLR2 macrophages and #15 IL1B monocytes, respectively. What message does this convey (Figure 4 panels C-F)

e. It is stated that TAM infiltration is associated with poor prognosis after R-CHOP therapy, did the author evaluate this in the non-relapsed versus relapsed DLBCL comparison ?

5. I am a little skeptical to the use of FFPE tissues for high throughput transcriptomics analysis, as formalin highly impacts RNAs.

6.a. It is not obvious that the resulting macrophage gene expression signatures could be used in routine practice. At this moment in time, there are a plethora of signatures (Imaging, DNA-, RNA-, and protein-based) that are all defining inferior prognosis in DLBCL. Some of the MacroSigs defined in this article are in fact only correlated with already available prognostic markers such as COO and genetic profiles, but do not add to these existing classifications. For example, the association between MacroSig#7 (no relapse) and EZB subtype (associated with good prognosis) is predictable. So is the association between MacroSig#8 (relapsed DLBCLs) and MCD subtype (associated with poor prognosis).

b. A multi-variate analysis is necessary to eliminate correlation with known prognostic markers that are used in routine practice. The most obvious aspect would be an association to TP53 and/or MYC/BCL2 double or single hit, which has been linked to inferior prognosis and is reported to be enriched in relapsed DLBCLs (as their signature).

7. Using spatial transcriptomics for routine practice would be difficult because of a high cost, but also due to the difficulty to analyze data locally. Do the authors plan to provide an online available tool for analyzing data ?

8. There is only a few novel aspects about tumor microenvironment biological features. R-CHOP immunochemotherapy is still the standard of care, but a number of new treatment combinations are also available, together with novel agents and immunotherapies. In terms of therapeutic approach for treating underlying tumor microenvironment, is there any new biological insight that can be used to pave the way for new DLBCL treatments/combinations? Can the authors expand on "macrophage-centered treatments"?

9. The authors delineated macrophage gene expression signatures potentially indicative of survival, then prone to guide treatment decisions. These results were generated from a single-center patients selection process. In this context, the confirmation of these results on an independent validation cohort is required.

Minor:

1. The raws are not numbered.

2. A little more effort in explaining the results should be made in the figures. The comments are too short to fully catch the results without reading the corresponding section.

3. The figures and their organization : Figure 1 and 2 for example, are made many (up to 10) (A-J) panels. A figure is something the reader should easily read to access the main information. I don't think the 10 panels of figure 2 can fit in a single printed page. Figures 6 and 7 : there are too many panels.

4. In Figure 4 comments there are up to 8 panels (A-H). However, the corresponding figure only displays 6 panels (A-F).

5. Figure 5 : may the authors explain how the gene ratios were calculated ?

6. Discussion section : CCO classifications  COO classifications.

Reviewer #2 (Remarks to the Author): Expert in spatial transcriptomics and tumour microenvironment

In this work, Liu et al. apply digital spatial profiling, a form of spatial transcriptomics, to diffuse large B-cell lymphoma samples, extracting gene expression data for 3 immune cell populations. They use the DSP data to identify 8 “MacroSigs” corresponding to various spatial regions and patient survival, suggesting that macrophages take on different cellular identities depending on their surroundings, and that these identities can contribute to patient outcomes.

The work is technically impressive and is a nice use case of the DSP technology, and the manuscript is written in a clear and easy to follow manner. While the biology suggested by the transcriptomics data is interesting, novel and has clear clinical relevance, the manuscript reads as a bit disjointed. The data itself is somewhat correlative/descriptive, and lacking any mechanistic experiments to validate the conclusions suggested by the transcriptomics data. The lack of corresponding scRNA-seq data makes it impossible to assess whether the MacroSigs themselves represent true macrophage cell states at the single-cell level.

In my opinion, several major changes are needed before the manuscript is suitable for publication in Nature Communications. A major weakness is the lack of detail describing the methods, particularly in how the MacroSigs were defined. This makes it difficult for me to be convinced that the MacroSigs were defined correctly and in a stringent manner, such that they truly represent novel macrophage populations. The fact that the MacroSigs not do map well onto existing macrophage scRNA-seq atlases (i.e MoMac) also raises concern that this may not in fact be due to the MacroSigs representing novel macrophage populations (as the authors suggest), but that the MacroSigs were sloppily defined (i.e too many genes, not stringent enough cutoffs). As it is likely that future researchers will use these signatures for understanding macrophage populations present in their own data, it is of critical importance to the field that (a) the MacroSigs are defined in a stringent way, and (b) the methods used to identify the MacroSigs are clearly described. Expanding both the main text and supplementary methods sections and modifying the figures as suggested below will help to alleviate these concerns.

Major comments:

1. In general, the Methods section is very light on details, making it difficult for the reader to understand how the analyses were done. This is a critical issue for such an analysis-heavy paper. While the supplementary methods have more information, both sections are fairly vaguely written (e.g “GSEA analyses was performed with R software”). Please rewrite both the main text and supplementary methods sections with enough detail such that others are able to reproduce the results (R packages used, parameters, cutoffs, etc).
2. Related to the above point, more information needs to be provided as to how the MacroSigs were defined - it is barely described in the text or methods. Is there any overlap between the gene lists, and if so, how is overlap accounted for? Why must genes match “either” a p-value cutoff or a log2FC cutoff - this seems like a fairly lenient criterion (why not both a p-value and a log2FC cutoff?). Why was a log2FC of 0.58 chosen as the cutoff (seems low)? Why were 50 genes chosen? The supplemental methods suggest in some cases, less than 50 genes were identified - why not use a higher log2FC cutoff or lower p-value to get cleaner lists with fewer genes, that might map better onto preexisting datasets?
3. As the authors themselves suggest in the Discussion, pairing the DSP results with scRNA-seq would clarify the functional role of these macrophage populations in DLBCL, and allow for more sophisticated

analyses of interactions between the different subtypes (pseudotime analysis, modelling of ligand-receptor signaling, etc). A scRNA-seq experiment would also provide critical confirmation at the single-cell level that the different MacroSigs truly represent different macrophage subtypes.

4. The description of the data presented in Fig. 5 needs improvement and to be expanded - there is a huge amount of biology presented here that is almost glossed over, with passing references to many different tumor subtypes and cell states that a non-DLBCL expert will have no knowledge of. Furthermore, the survival data, while mentioned in the text, is not shown in the figure. I am not sure if so many dot plots are the best way to display this data. Perhaps the data could be plotted another way in which the corresponding survival data is also shown.

Minor comments:

1. Top of page 6: please describe which cell types CD68, CD3, and CD20 are labelling - it is written in the figure legend but not in the manuscript text. In general, it would be helpful for the reader if the corresponding cell type is mentioned every time a cell type marker is referred to.

2. Can the authors comment on the approximate size of each ROI and AOI? Approximately how many cells are captured per AOI? This is important to understand the resolution of the DSP data.

3. Please change the colours in Fig. 1B, bottom right panel - the 2 purple colours labelling the CD68 and CD20 populations are difficult to distinguish.

4. Fig. S2B: I am curious why the clustering of the DLBCL scRNA-seq data is so poor, with many overlapping populations that I would expect to be fairly separate in UMAP space based on gene expression. Can the authors comment on how they integrated these 10 different datasets? Including a UMAP in which the cells are labelled by original dataset would also be helpful to visualize how well the integration worked.

5. Fig. S2C,D: Diverging colour maps should not be used for unscaled data.

6. Fig. 2: how well can the authors define MacroSigs corresponding to LZ vs DZ when there are no statistically significant DEGs between the two groups? Same goes for Fig. 4B.

7. Figs. 2G-J and 3E-H: I am not sure what purpose this analysis serves, other than indicating that the MacroSigs poorly correspond to different macrophage subtypes. 50 genes are a lot to be plotting, so I'm not surprised that the authors see poor cell-type specific expression of the MacroSigs. This goes back to the issue of how the MacroSigs were defined - with so few details, it is hard for the reader to be convinced that the MacroSigs are a stringent representation of these cell populations.

8. I am not sure if Fig. 3A is needed - it is not very convincing (shows little difference in gene expression between the DLBCL and RLT AOIs) and the clustering is not great. Presumably the authors know which AOIs came from which tissue without the need for this analysis. Fig. 3C is much more convincing in showing changes in gene expression between the two groups.

9. Fig. 3: it would be helpful to include a heatmap with the expression of the genes corresponding to MacroSigs 5 and 6 plotted for DLBCL and RLT AOIs.

10. Fig. 5: It would be interesting to see how the MacroSigs map in UMAP space for the datasets in Fig. 5E-H (if these are in fact single cell datasets - this is not mentioned in the figure legend or the text).

11. Fig. 6: The authors map the MacroSigs onto bulk RNA-seq data from DLBCL patient samples, and state that the RNA-seq data likely contains macrophages since they are such a large component of the TME. What proportion of macrophages (relative to tumor) are found in these datasets? Can the authors infer the approximate proportion of macrophages based on macrophage gene expression in the bulk RNA-seq data?

12. The Discussion is too long (almost 5 pages) and should be cut down.

13. Supplementary tables should be included as Excel or CSV files - not PDFs.

14. The paper relies heavily on acronyms - it seems like overkill at times and makes it difficult to read. Page 12 itself must contain close to 20 different acronyms.

Reviewer #3 (Remarks to the Author): Expert in lymphomas, immunology, and macrophages

In this manuscript, entitled "Spatially-resolved transcriptomics reveal macrophage heterogeneity and prognostic significance in diffuse large B-cell lymphoma", Liu et al, characterized macrophages in DLBCL tissues by using a spatial transcriptome approach.

Recent single-cell RNA-sequencing studies have revealed heterogeneity of tumor-associated macrophages; however, spatial heterogeneity remains to be fully characterized. This manuscript defines macrophages in DLBCL tissues by a spatial transcriptomics approach. Specifically, the authors defined 8 spatially-derived macrophage signatures (MacSigs), which could clearly separate macrophages within normal reactive lymph nodes, as well as macrophages from different disease statuses (normal vs DLBCL vs relapsed DLBCL). Overall, the authors carefully analyzed data, and results in this manuscript contain high novelty and significance.

I have a few suggestions.

1. The authors defined eight spatially-derived macrophage signatures. To validate the results, the authors investigated clinical relevance (Cell-of-Origin and prognosis), using published DLBCL datasets. However, the authors need to validate key macrophage signatures by IHC staining. This would provide more clinical implications (i.e., clinic-pathological analysis in the real-world clinical setting).

#2. In addition to CD163, could the authors see differences in the expression levels of commonly used M2-related markers among 8 MacSigs (such as Arg-1, CD204, CD206, VEGF)? Given that DCs also express CD68, is there any possibility that one of these MacSigs (especially Light Zone) expresses DC marker genes?

#3. It is now recognized that macrophage phagocytosis checkpoint molecules are associated with resistance to rituximab. Could the authors see differential expression levels of molecules (SIRPa, Siglec-10, LILRB1, and PD-1) among 8 MacSigs?

We thank the reviewers for their time to read through and analyse our work. We are very appreciative of the constructive nature of their feedback and have extensively revised the manuscript and figures based on it. The comments and suggestions have greatly strengthened the manuscript, and we hope the revised version and the point-by-point replies below (in blue) will address their concerns.

Responses to reviewer comments:

Reviewer #1 (Remarks to the Author): Expert in DLBCL genomics and clinical reserach

In this work, Liu and colleagues use spatial transcriptomics to explore tumor microenvironment in DLBCL, tumor-associated macrophages in particular. The authors conducted a comparative mapping of macrophages gene expression patterns within FFPE tissues of reactive lymphoid tissues (RLT; n=17) and diffuse large B-cell lymphoma (DLBCL; n=47). The digital spatial profiling with whole transcriptome performed on CD68+ cells in different conditions provided differentially expressed genes (DEG) that were projected to a public single cell RNA-seq macrophage atlas, leading to the definition of 8 spatially-derived macrophage gene expression signatures (named MacroSig 1 to 8). MacroSig#1 corresponds to germinal center (GC), #2 to interfollicular (IF) region, #3 to light zone of GC, #4 to dark zone of GC, #5 to reactive lymphoid tissue (all GC areas), #6 to DLBCL, #7 to non-relapsed DLBCL and #8 to relapsed DLBCLs. Some of them are associated with survival when applied on publicly available DLBCL datasets.

This is an interesting topic, explored with innovative tools. However, in its current form, some aspects of this article are not easily accessible to the readership. The methods used to define the different macrophages gene expression signatures are not fully convincing and should be better explained. Only a few insights into biological features are provided and the underlying DLBCL tumor microenvironment is not extensively explored. I am not sure that the prognostic markers provided in this work will emerge from the plethora of already available DLBCL predictive markers.

Response: We are glad that the reviewer found the work interesting and innovative. We are especially grateful for them highlighting aspects that were not clear to a reader and have extensively revised our writing based on their feedback. Overall, in this paper we now have clarified our focus to highlight and clearly describe spatially resolved gene expression signatures arising from distinct macrophage populations in reactive/malignant lymphoid tissues, and how infiltration by specific subtypes of macrophages may correlate with clinicopathological features and survival in DLBCL. We agree that the biological mechanisms underpinning the negative prognostic effect of dark zone macrophages in the microenvironment will be very interesting, and hope to address this in a subsequent paper.

Major issues:

Comments: 1.a. the results are supported by small sample sizes, and derived statistics

for determining DEG between the different groups seems insufficiently stringent for manipulating such a high amount of data. The authors used moderated t-test, Fisher exact test, with a significant p-value set at 0.5. To be selected as a differentially expressed between two groups, a gene must meet at least one of the 2 following criteria: (i) adjusted p-value $< .5$ after Benjamini-Hochberg correction, (ii) an absolute \log_2 (fold change) > 0.58 . So the fold change is sufficient for selection without passing a statistical test.

Response: We thank the reviewer for highlighting this important area for improvement and apologise for the lack of clarity and detail in our initial submission regarding statistical definitions criteria for the macrophage signatures (MacroSigs). We have taken on board the reviewers' comment that the numbers of regions/cases were small thereby limiting statistically robust definitions of these gene expression signatures. To ensure that we only define macrophage signatures based on genes that pass both the fold-change and adjusted P value criteria, we performed new additional DSP experiments to expand our samples size and strengthen the comparative analyses (A new total of 702 regions; with 24 reactive lymphoid tissues (RLTs) and 87 DLBCL samples; details in Supplementary Fig. 1A). We evaluated differentially expressed genes (DEGs) of 4 comparisons (GC versus IF; LZ versus DZ; RLT versus DLBCL; No relapse versus Relapse) based on both criteria: a. Benjamini-Hochberg adjusted P value < 0.05 ; b. Absolute \log_2 fold change ($|\log_2FC|$) > 0.58). MacroSigs1-6 fulfilled both requirements, thereby demonstrating their stringency and reliability. The only exception was the comparison between MacroSig7 (No relapse) and MacroSig8 (Relapse) as no significant DEGs were present between No relapse/Relapsed-DLBCL macrophages after P -value correction. We have accordingly removed these two signatures from our revised manuscript. We have also extensively revised the writing about MacroSig derivation in the Methods section (Page 25, line 16-22; Page 26, line 1-4) and main text of the manuscript (Page 9, line 15-20; Page 11, line 2-4) to make it clearer for readers.

Comments: b. I am a little skeptical about classifying gene expression levels into 3 groups (high, intermediate and low) and compare survival only between the 2 extreme groups for subsequent survival analyses. I did not find the classification threshold for these three groups, even in the supplementary methods. Why removing the intermediate group from the analysis?

Response: We chose a 3 group tertiles split based on the gaussian distribution of patients based on MacroSig scores (Fig. A below). In a bimodal distribution it would have been clear to divide the population into two groups, but with this gaussian distribution, the intermediate tertile consists of patients with an intermediate score that do not clearly fall into "high" or "low" signature expression groups. We therefore compared the clearly high and low tertile groups, in an approach that is indeed widely used for survival analysis [1]. We have added more details about the tertiles and the rationale in the Methods section of the main manuscript (Page 27, line 13-21; Page 28, line 1-22; Page 29, line 1-3). Importantly, as Kaplan-Meier analysis requires dichotomisation of the population into groups, we also analysed the survival effect of

MacroSig scores as a continuous variable. These results (Fig. B-C below) also show a statistically significant impact on survival for both MacroSig4 (DZ) and MacroSig6 (DLBCL), attesting to the overall validity of our findings. We have not currently included the continuous variable analysis in the manuscript but are open to doing so.

[1] Raman P, et al. A comparison of survival analysis methods for cancer gene expression RNA-Sequencing data. *Cancer Genet.* 2019 Jun;235-236:1-12. doi: 10.1016/j.cancergen.2019.04.004. Epub 2019 Apr 12. PMID: 31296308.

Figure legends: (A) The distribution of the DZ-LZ scores in the Sha et al. dataset. The weighted total expression score was considered as the continuous explanatory variable. (B) Univariate Cox proportional hazards model analysis for MacroSig3 (LZ) and MacroSig4 (DZ) in eight publicly available DLBCL datasets using the continuous explanatory variables. (C) Univariate Cox proportional hazards model analysis for MacroSig5 (RLT) and MacroSig6 (DLBCL) in eight publicly available DLBCL datasets using the continuous explanatory variables.

Comments: 2.F a. Little is told about sample selection: how was the sample selection conducted (supplemental methods only mention "random selection")? Among DLBCLs, did the authors include de novo DLBCLs only or are there also transformed DLBCLs?

Response: We thank the reviewer for highlighting this and apologise for the oversight-

the term random selection was inaccurately used. We included cases of de novo DLBCL diagnosed between 2010 and 2017 at the National University Hospital Singapore, which had adequate archival FFPE tissue for the extraction of cores into tissue microarray format, in a study approved by the Singapore NHG Domain Specific Review Board B study protocol (2015/00176). The DSP analyses were performed on the samples on tissue microarrays from patients who had completed 6 cycles of R-CHOP. We have clarified this in the methods section (Page 20, line 10-22; Page 21, line 1-5) and have included a consort diagram (see below, and Supplementary Fig. 8).

Figure legend: Patients' selection flowchart. Cases of de novo DLBCL diagnosed between 2010 and 2017 at the National University Hospital Singapore were included in this study. The criteria for selecting patients for DSP analysis is shown.

Comments: b.How did the author construct the non-relapsed DLBCL group with a ~6-year median follow-up? A subset of DLBCL relapses emerge later. What is the method used to assess relapse? Did they all undergo FDG-PET/CT? Were DLBCL patients sub-classified into transcriptomic-based ABC or GCB categories? Or in the genetic-based DLBCL subsets more recently identified?

Response: The vast majority (>90%) of DLBCL relapses occur in the first two years of treatment after R-CHOP [2], with the risk largely plateauing after 4 years. Therefore we felt it reasonable for this initial analysis to define “no relapse” as patients who do not relapse within a 3 year time frame. Our clinical protocols mandate FDG PET assessments of relapse. In keeping with current clinical guidelines worldwide (<https://meridian.allenpress.com/aplm/article/145/3/269/447620/Laboratory-Workup-of-Lymphoma-in-Adults-Guideline>), our samples do not have routine transcriptomic or genetic subtyping of DLBCL at initial diagnosis. However, we have analysed the relationship of our signatures to gene expression-based subtypes and genetic subtypes using larger DLBCL datasets from seminal work in the field [3-10]. These details are also now included in the methods section for clarity (Page 21, line 6-22; Page 29, line 4-14).

[2] Ekberg S, et al. Patient trajectories after diagnosis of diffuse large B-cell lymphoma—a multistate modelling approach to estimate the chance of lasting remission. *Br J Cancer*. 2022 Nov;127(9):1642-1649. doi: 10.1038/s41416-022-01931-2. Epub 2022 Aug 23. PMID: 35999271; PMCID: PMC959649

[3] Schmitz R, et al. Genetics and Pathogenesis of Diffuse Large B-Cell Lymphoma. *N Engl J Med*. 2018 Apr 12;378(15):1396-1407. doi: 10.1056/NEJMoa1801445. PMID: 29641966; PMCID: PMC6010183.

[4] Lacy SE, et al. Targeted sequencing in DLBCL, molecular subtypes, and outcomes: a Haematological Malignancy Research Network report. *Blood*. 2020 May 14;135(20):1759-1771. doi: 10.1182/blood.2019003535. PMID: 32187361; PMCID: PMC7259825.

[5] Chapuy B, et al. Molecular subtypes of diffuse large B cell lymphoma are associated with distinct pathogenic mechanisms and outcomes. *Nat Med*. 2018 May;24(5):679-690. doi: 10.1038/s41591-018-0016-8. Epub 2018 Apr 30. Erratum in: *Nat Med*. 2018 Aug;24(8):1292. Erratum in: *Nat Med*. 2018 Aug;24(8):1290-1291. PMID: 29713087; PMCID: PMC6613387.

[6] Lenz G, et al. Stromal gene signatures in large-B-cell lymphomas. *N Engl J Med*. 2008 Nov 27;359(22):2313-23. doi: 10.1056/NEJMoa0802885. PMID: 19038878; PMCID: PMC9103713.

[7] Sha C, et al. Molecular High-Grade B-Cell Lymphoma: Defining a Poor-Risk Group That Requires Different Approaches to Therapy. *J Clin Oncol*. 2019 Jan 20;37(3):202-212. doi: 10.1200/JCO.18.01314. Epub 2018 Dec 3. Erratum in: *J Clin Oncol*. 2019 Apr 20;37(12):1035. PMID: 30523719; PMCID: PMC6338391.

[8] Visco C, et al. Comprehensive gene expression profiling and immunohistochemical studies support application of immunophenotypic algorithm for molecular subtype classification in diffuse large B-cell lymphoma: a report from the International DLBCL

Rituximab-CHOP Consortium Program Study. *Leukemia*. 2012 Sep;26(9):2103-13. doi: 10.1038/leu.2012.83. Epub 2012 Mar 22. Erratum in: *Leukemia*. 2014 Apr;28(4):980. PMID: 22437443; PMCID: PMC3637886.

[9] Dubois S, et al. Biological and Clinical Relevance of Associated Genomic Alterations in MYD88 L265P and non-L265P-Mutated Diffuse Large B-Cell Lymphoma: Analysis of 361 Cases. *Clin Cancer Res*. 2017 May 1;23(9):2232-2244. doi: 10.1158/1078-0432.CCR-16-1922. Epub 2016 Dec 6. PMID: 27923841.

[10] Reddy A, et al. Genetic and Functional Drivers of Diffuse Large B Cell Lymphoma. *Cell*. 2017 Oct 5;171(2):481-494.e15. doi: 10.1016/j.cell.2017.09.027. PMID: 28985567; PMCID: PMC5659841.

Comments: 3. a. Regarding tissue sample selection, did the authors use only lymph node biopsies for DLBCLs? Since the authors worked on bulk tissues, it is of major importance to accurately determine the relative proportion of tumor cells and microenvironment cells. At least, was the tumor purity assessed (large cells component)? Did the authors perform a pathology review to confirm the diagnosis?

Response: Yes, co-authors Dr. Ng Siok Bian and Dr. Susan Hue are haematopathologists who selected tumour-rich regions (avoiding necrosis/ adipose tissue) for construction of our DLBCL tissue microarrays at NUH. The biopsies included lymph node samples and some extra-nodal tissue samples- details are now included in a Supplementary Table 5. While we performed a CD20 (B-cell) mask analysis by DSP for comparison, in this study we focus on the readout of the CD68 mask (which is independent of relative proportions of malignant and non-malignant B-cells). Importantly, we show that the gene expression signatures from these CD68 masks are very distinct from those in the CD20 masks, and also project to the myeloid-macrophage cluster in a large single-cell RNA seq dataset (also see response to comment 3 from reviewer 2), highlighting the technical accuracy of the DSP approach.

Comments: b. It is stated that "macrophages are abundant cells in the microenvironment of DLBCL". Was an objective test performed to evaluate it properly for each sample (at least to ensure that this proportion is equivalent in all tested samples)?

Response: This was a statement in our introduction to the study, and we have added the following references in support:

[11] Li YL, et al. Tumor-associated macrophages predict prognosis in diffuse large B-cell lymphoma and correlation with peripheral absolute monocyte count. *BMC Cancer*. 2019 Nov 6;19(1):1049. doi: 10.1186/s12885-019-6208-x. PMID: 31694577; PMCID: PMC6836332.

[12] Riihijärvi S, et al. Prognostic influence of macrophages in patients with diffuse large B-cell lymphoma: a correlative study from a Nordic phase II trial. *Haematologica*. 2015 Feb;100(2):238-45. doi: 10.3324/haematol.2014.113472. Epub 2014 Nov 7. PMID: 25381134; PMCID: PMC4803141.

Comments: c.Figure 1D: it seems that CD68 is not fully macrophage-specific, as there is a CD68-originating component in the T-cells and in the B-cells signature, suggesting that a portion of this CD68 signature is wrongly attributed to these cell types. Moreover, figure 2C: E2F (a B-cell transcription factor) targets were the most enriched pathway in GC macrophages. Do the authors consider that CD68 staining is specific enough to accurately and specifically select macrophages within tissues?

Response: We thank the reviewer for raising this as an important point to address. The NanoString GeoMx platform does not work at true single cell resolution, but using mask delineated by immunofluorescent staining allows an enrichment for transcripts from cell types of interest- here CD68 being used for macrophages [13-14]. While it is possible that some of the CD68 signature contains genes arising from B and T cells due to the optical resolution of the technique, we performed a few analyses to cross check this issue. First, we used published single-cell RNAseq datasets to validate that a set of genes that show high specificity for B-cells (MS4A1, CD79A, CD79B, CD19, and PAX5), T-cells (CD3D, CD3E, UBASH3A, CD2, and TRBC2) and Macrophages (CD68, CD163, CSF1R, and FCGR1A) project to their respective cell types. Then we show in Fig. 1D-E that these “mini-signatures” are enriched within the masks of our DSP experiment. While there is some capture of these macrophage and T-cell genes within the B-cell mask (albeit much lower than B-cell genes and expected to some extent as the CD20 B-cell mask is the most abundant/ broad of the three masks used), the key comparison here is that these canonical and highly expressed B-cell and T-cell genes are not captured in the CD68 mask. This suggests that our CD68 based signatures are likely to primarily consist of macrophage derived genes. The E2F family of transcription factors are components of the DREAM complex and critical to cell division in a variety of tissues- they are not B-cell specific, and are important for cells of myeloid lineage as well [15]. On the choice of CD68 as a mask, CD68 is a well described marker for cells of macrophage lineage [13-14]. While it is also expressed in monocytes/ dendritic cells, it is not expected to be expressed in B and T cells. We intentionally chose a broad marker for the DSP WTA experiment to capture as many subtypes of cells of the monocyte-macrophage lineage for subsequent transcriptomic characterization. We have clarified on these points in the writing as well (Page 8, 18-20; Page 15, 17-18), and included a statement in the discussion on the limitations of DSP and CD68 as a macrophage mask (Page 18, line 21-22; Page 19, line 1-19).

Importantly, we have now also projected all genes from our MacroSigs in a scRNA-seq DLBCL dataset from Ye et al. [16] (N = 17, comprising transcriptomes of 94,324 cells) using a module score. We show that most MacroSigs (except for GC/RLT which share several cell-cycle/ proliferation genes with the GC/RLT B-cell signature from our DSP experiment) showed a clear enrichment and higher module scores in the Monocytes/Macrophages population over the B-cell population (Fig. 4D and Supplementary Fig. 5). Finally, there is little overlap between the actual genes in the CD68-derived signatures and CD20-derived signatures of the germinal centre dark zone (Fig. 7B). Thus, we believe that despite the optical limitations of the DSP technique in terms of single-cell resolution, our signatures based on aggregate data within these masks are largely representative of the intended cell type.

[13] Moutafi M, et al. Discovery of Biomarkers of Resistance to Immune Checkpoint Blockade in NSCLC Using High-Plex Digital Spatial Profiling. *J Thorac Oncol.* 2022 Aug;17(8):991-1001. doi: 10.1016/j.jtho.2022.04.009. Epub 2022 Apr 28. PMID: 35490853; PMCID: PMC9356986.

[14] Liu J, et al. Transcriptional and Immune Landscape of Cardiac Sarcoidosis. *Circ Res.* 2022 Sep 30;131(8):654-669. doi: 10.1161/CIRCRESAHA.121.320449. Epub 2022 Sep 16. PMID: 36111531; PMCID: PMC9514756.

[15] Trikha P, et al. E2f1-3 are critical for myeloid development. *J Biol Chem.* 2011 Feb 11;286(6):4783-95. doi: 10.1074/jbc.M110.182733. Epub 2010 Nov 28. PMID: 21115501; PMCID: PMC3039366.

[16] Ye X, et al. A single-cell atlas of diffuse large B cell lymphoma. *Cell Rep.* 2022 Apr 19;39(3):110713. doi: 10.1016/j.celrep.2022.110713. PMID: 35443163.

Comments: d. Last, the lymph node architecture is commonly highly disrupted in DLBCL, how do the authors delineate CD68+ areas in this context, with a widely dominant tumor cell content?

Response: The GeoMx technique works through UV-based cleavage and microfluidic capture of RNA probes from “areas of interest” (AOIs) defined either geographically, or through a secondary stain. In our experiments we used the secondary immunostaining approach (described as “masks”). Despite these cells being closely admixed with tumour cells in the DLBCL microenvironment (see Fig. 1 for a representative image), the DSP system is able to capture and enrich for genes expressed from cells expressing CD68. The optics and fluidics of the system do not offer true single cell resolution, but overall our data (and others in a variety of different tissue contexts [17-18]) suggest good cell type enrichment even within a complex microenvironment using the DSP method (Fig. 1D-E).

[17] Jerby-Arnon L, et al. Opposing immune and genetic mechanisms shape oncogenic programs in synovial sarcoma. *Nat Med.* 2021 Feb;27(2):289-300. doi: 10.1038/s41591-020-01212-6. Epub 2021 Jan 25. PMID: 33495604; PMCID: PMC8817899.

[18] Song X, et al. Spatial multi-omics revealed the impact of tumor ecosystem heterogeneity on immunotherapy efficacy in patients with advanced non-small cell lung cancer treated with bispecific antibody. *J Immunother Cancer.* 2023 Feb;11(2):e006234. doi: 10.1136/jitc-2022-006234. PMID: 36854570; PMCID: PMC9980352.

Comments: 4.a. In devising their macrophage gene-expression signatures, the authors used comparative transcriptomics. The design is not clear, and some groups are overlapping: first the authors compared GC versus interfollicular area (MacroSig 1 and 2), then the light zone versus dark zone (2 components of the GC; MacroSig 3 and 4), then RLT (ie light zone, dark zone, GC and interfollicular area) versus DLBCL (MacroSig 5 and 6) and ultimately non-relapsed versus relapsed DLBCL (MacroSig 7 and 8). The authors should better explain the design used to define these signatures, and

how the associated statistical methods support these results.

Response: We thank the reviewer for highlighting this point for clarification. Given the highly compartmentalised roles between different regions of lymphoid tissues, we were interested in whether macrophages within these areas showed unique gene expression patterns. We therefore compared CD68 cells in the germinal centre [19-20] with interfollicular regions, and then between the dark zone and the light zone within the germinal centre. These comparisons comprise the first 4 MacroSigs. In our current revision, we only show one other comparison, and that is of tumour vs reactive tissue (GC was chosen as the comparator given that DLBCL is thought to originate from different functional stages of GC B cells. With the increased sample size in the revision, all the above MacroSigs are now statistically significant. In the revised manuscript, we have explained the design, derivation and comparisons of the MacroSigs in the Methods section (Page 25, line 16-22; Page 26, line 1-4).

[19] Young C, Brink R. The unique biology of germinal center B cells. *Immunity*. 2021 Aug 10;54(8):1652-1664. doi: 10.1016/j.immuni.2021.07.015. PMID: 34380063.

[20] Koopman G, Pals ST. Cellular interactions in the germinal center: role of adhesion receptors and significance for the pathogenesis of AIDS and malignant lymphoma. *Immunol Rev*. 1992 Apr;126:21-45. doi: 10.1111/j.1600-065x.1992.tb00629.x. PMID: 1597319.

Comments: b. Results, first paragraph: in RLTs' GC, light zone- and dark zone-originating macrophages present with no statistically significant differentially expressed genes (DEG). Then how could the authors derive MacroSigs from this? Later in the results section, and also in the summary: DZ MacroSig (#4) is associated with shorter survival while LZ MacroSig (#3) is not. How do you explain this? The same applies to MacroSig#7 and #8, as no DEG passed the significance threshold after p-value correction.

Response: We thank the reviewer for directing our attention to this important consideration. We agree that it would be appropriate to only define MacroSigs with differentially expressed genes based on statistical significance (by adjusted *P*-values). We previously described 4 pairs of MacroSigs: MacroSig1/2 (GC/IF), MacroSig3/4 (DZ/LZ), MacroSig5/6 (RLT/DLBCL), and MacroSig7/8 (No relapse/Relapse). For MacroSig1/2 (GC/IF) and MacroSig5/6 (RLT/DLBCL), we defined the MacroSigs based on the both criteria: a. Benjamini-Hochberg adjusted *P*-value < 0.05; b. Absolute log₂ fold change ($|\log_2FC|$) > 0.58. For the MacroSig3/4 (LZ/DZ) and MacroSig7/8 (No relapse/Relapse), many genes were distinctly expressed between LZ/DZ and non-relapsed/relapsed macrophages, but were not individually statistically significant after *P*-value correction. Thus, we defined these MacroSigs based on the of criteria: a. unadjusted *P*-value < 0.05 (unadjusted *P*-value < 0.05 as there were no significant DEGs in these comparisons after Benjamini-Hochberg correction); b. Absolute log₂ fold change ($|\log_2FC|$) > 0.58.

Currently, to define our MacroSigs in a more stringent and robust fashion, we increased the sample size with new additional DSP experiments (207 CD68+ regions from

different spatial niches of 24 reactive lymphoid tissues; 112 CD68+ regions from 87 DLBCL patients; As shown in Supplementary Fig. 1A). We subsequently derived the MacroSigs from the DEGs based on the both criteria: a. Benjamini-Hochberg adjusted P -value < 0.05 ; b. Absolute \log_2 fold change ($|\log_2FC| > 0.58$). In the revised derivation, MacroSigs1-6 fully achieved both requirements. However, MacroSig7 (No relapse) and MacroSig8 (Relapse) did not as there were no significant DEGs between no-relapsed/relapsed-DLBCL macrophages after P -value correction. Therefore, we removed the MacroSig7-8 from the revised manuscript. We have thoroughly modified the writing about the derivation of MacroSigs in the Methods section (Page 25, line 16-22; Page 26, 1-4) and main text of the manuscript (Page 9, line 15-20; Page 11, line 2-4) to make it clearer for the reader to understand.

Comments: c. When projected on publicly available datasets, MacroSig#6 (derived from RLT (GC area) versus DLBCL comparison) was enriched in ABC subtype DLBCLs. It seems predictable, as genes highly expressed in RLT's GC were removed from MacroSig#6.

Response: The classification of DLBCL into cell-of-origin (COO) categories is based on gene expression patterns reminiscent of germinal center B cells (GCB) and activated B cells (ABC). However, our MacroSigs were derived from CD68 expressing cells, and the GC MacroSig has a clearly distinct set of genes from the GC B-cells. The association of MacroSigs with specific COOs is therefore interesting, and may suggest that DLBCL COO specification may co-evolve with unique macrophage subgroups and that macrophages expressing specific transcriptional programs dictated by their topographic localization (e.g., residing in the GC) may differently enrich lymphomas with diverse COO. This hypothesis, however, requires further biological validation and we have only alluded to it in the discussion (Page 16, line 20-22; Page 17, line 1-8).

Comments: d. MOMac-VERSE: this tool was used to project MacroSigs on a gene expression-derived map of 17 annotated monocytes/macrophages subtypes. But most of the times, this does not lead to a clear labelling, except for MacroSig#7, and MacroSig#8, which seem to cluster within #2 HES1/FOLR2 macrophages and #15 IL1B monocytes, respectively. What message does this convey (Figure 4 panels C-F)

Response: We thank the referee for highlighting this point and have elaborated on it in the manuscript. While the clear projection of some signatures onto previously defined cell types in MoMac-VERSE is of interest, the absence of this does not imply lack of significance, as the MoMac-VERSE is generated on single-cell data that is not spatially resolved. For example, we see very clear transcriptomic differences between GC and IF macrophages, and IF macrophages in DLBCL correlate with unclassified COO cases. However, the IF macrophage signature does not cluster with known cell types in MoMac-VERSE, suggesting that there may be additional cell types that need to be incorporated into newer versions of the MoMac-VERSE, or that transcriptional clustering with spatial information may follow different hierarchies from single cell suspension-based analyses. This will be a topic of future study in our and our collaborators lab.

Comments: e. It is stated that TAM infiltration is associated with poor prognosis after R-CHOP therapy, did the author evaluate this in the non-relapsed versus relapsed DLBCL comparison?

Response: We have added the following references in support of this general statement [21-23]. In our current manuscript we have removed the non-relapsed vs relapsed macrophage signatures due to the lack of statistical significance after *P*-value correction. We hope to explore this topic (the nature of macrophages in patients with relapsed vs non-relapsed DLBCL) in a dedicated subsequent study with additional samples from multiple cohorts.

[21] Zeng H, et al. CD163+ Tumor Associated Macrophages Predict Inferior Outcome in Patients with Diffuse Large B-Cell Lymphoma Treated with R-CHOP. *Blood*. 2015 126 (23): 5023. doi: 10.1182/blood.V126.23.5023.5023.

[22] Li YL, et al. Tumor-associated macrophages predict prognosis in diffuse large B-cell lymphoma and correlation with peripheral absolute monocyte count. *BMC Cancer*. 2019 Nov 6;19(1):1049. doi: 10.1186/s12885-019-6208-x. PMID: 31694577; PMCID: PMC6836332.

[23] Croci GA, et al. SPARC-positive macrophages are the superior prognostic factor in the microenvironment of diffuse large B-cell lymphoma and independent of MYC rearrangement and double-/triple-hit status. *Ann Oncol*. 2021 Nov;32(11):1400-1409. doi: 10.1016/j.annonc.2021.08.1991. Epub 2021 Aug 24. PMID: 34438040.

Comments: 5. I am a little skeptical to the use of FFPE tissues for high throughput transcriptomics analysis, as formalin highly impacts RNAs.

Response: While formalin fixation affects RNA fragmentation and integrity, it does not cause complete degradation of the RNA transcripts, which show average fragment size of around 150bp in well preserved archival samples. The GeoMx approach is based on the complementation of probes with a probe binding domain of 35-50 bases and is well optimised for FFPE samples. There are several papers which now attest to the robustness of the DSP approach in FFPE tissue [17, 24-25]. Moreover we (CT/ADJ) have previously successfully applied NanoString DSP to FFPE samples of reactive lymphoid tissue [26].

[17] Jerby-Arnon L, et al. Opposing immune and genetic mechanisms shape oncogenic programs in synovial sarcoma. *Nat Med*. 2021 Feb;27(2):289-300. doi: 10.1038/s41591-020-01212-6. Epub 2021 Jan 25. PMID: 33495604; PMCID: PMC8817899.

[24] Roelands J, et al. Transcriptomic and immunophenotypic profiling reveals molecular and immunological hallmarks of colorectal cancer tumorigenesis. *Gut*. 2023 Jul;72(7):1326-1339. doi: 10.1136/gutjnl-2022-327608. Epub 2022 Nov 28. PMID: 36442992; PMCID: PMC10314051.

[25] Carpenter ES, et al. Analysis of Donor Pancreata Defines the Transcriptomic Signature and Microenvironment of Early Neoplastic Lesions. *Cancer Discov*. 2023 Jun

2;13(6):1324-1345. doi: 10.1158/2159-8290.CD-23-0013. PMID: 37021392; PMCID: PMC10236159.

[26] Tripodo C, et al. A Spatially Resolved Dark- versus Light-Zone Microenvironment Signature Subdivides Germinal Center-Related Aggressive B Cell Lymphomas. *iScience*. 2020 Sep 16;23(10):101562. doi: 10.1016/j.isci.2020.101562. PMID: 33083730; PMCID: PMC7522121.

Comments: 6.a. It is not obvious that the resulting macrophage gene expression signatures could be used in routine practice. At this moment in time, there are a plethora of signatures (Imaging, DNA-, RNA-, and protein-based) that are all defining inferior prognosis in DLBCL. Some of the MacroSigs defined in this article are in fact only correlated with already available prognostic markers such as COO and genetic profiles, but do not add to these existing classifications. For example, the association between MacroSig#7 (no relapse) and EZB subtype (associated with good prognosis) is predictable. So is the association between MacroSig#8 (relapsed DLBCLs) and MCD subtype (associated with poor prognosis).

Response: We wish to clarify that our aim in this work was not to define a new predictor of DLBCL outcome to be used in routine practice. Our studies are not designed or powered to do so. Our paper focuses on descriptions of hitherto unappreciated distinctions in gene expression between macrophages in different lymphoid spatial locations and physiological/ pathological states. Nonetheless, the associations of these spatially defined MacroSigs with known clinicopathological states in DLBCL (eg. COO subclassification, genetic profile, poor survival) highlights that information from lymphoma-associated macrophages may complement information from the malignant B-cell to eventually better refine patient stratification. Our findings are considered hypothesis-generating in this context, to be formally compared to other DLBCL predictors in prospective studies. We thank the reviewer for highlighting this point and have clarified this explicitly in the text (Page 17, line 10-12)

Comments: b. A multi-variate analysis is necessary to eliminate correlation with known prognostic markers that are used in routine practice. The most obvious aspect would be an association to TP53 and/or MYC/BCL2 double or single hit, which has been linked to inferior prognosis and is reported to be enriched in relapsed DLBCLs (as their signature).

Response: We thank the reviewer for highlighting this. In our revised version, we focus on MacroSig4 (DZ), which associates with poor prognosis in 7 out of 8 independent GEP datasets. We agree that a multivariate analysis would be valuable, however the most common prognosticators in routine clinical practice are the IPI score and double hit lymphoma (DHL) status. We were able to perform multivariate analysis (Page 13, line 20-22; Page 14, 1-3) adjusted for the IPI score and DHL in 5 out of 6 datasets (The IPI scores are not available in 2 datasets; DHL is only available in 2 datasets), and show that MacroSig4 (DZ) is indeed an independent high risk prognostic factor for survival of DLBCL patients. As we cannot exclude the association of specific molecular features such as TP53 and DHL with specific macrophage infiltration patterns (which will need

further studies in larger datasets), we believe that the IPI based multivariate analysis is most appropriate at this stage (to exclude clinical reasons for poor outcome).

Comments: 7. Using spatial transcriptomics for routine practice would be difficult because of a high cost, but also due to the difficulty to analyze data locally. Do the authors plan to provide an online available tool for analyzing data?

Response: We thank the reviewer for their interest. We have used the NanoString GeoMx DSP platform, which has community pipelines for easy access to data analysis solutions using R or python:

[“https://github.com/Nanostring-Biostats/GeomxTools](https://github.com/Nanostring-Biostats/GeomxTools)

<https://github.com/Nanostring-Biostats/GeoMxWorkflows”>

Additionally, there are also multiple community released pipelines available that enable democratized access to spatial data analysis at an open sourced freeware access as follows:

[“https://github.com/DavisLaboratory/standR”](https://github.com/DavisLaboratory/standR”)

We look forward to additional cloud-based informatics portals that enables global collaboration and data sharing.

Comments: 8. There is only a few novel aspects about tumor microenvironment biological features. R-CHOP immunochemotherapy is still the standard of care, but a number of new treatment combinations are also available, together with novel agents and immunotherapies. In terms of therapeutic approach for treating underlying tumor microenvironment, is there any new biological insight that can be used to pave the way for new DLBCL treatments/combinations? Can the authors expand on "macrophage-centered treatments"?

Response: Thank you for your comments. For example, one of the negatively prognostic MacroSigs in our study was MacroSig6 (DLBCL), which mapped to IL4I1+ macrophages with pro-tumorigenic effects [27]. As IL4I1+ macrophages are known to depend on AhR dependent signalling, such a finding could suggest the possibility of combining clinical grade AhR inhibitors with chemotherapy in DLBCL. This however requires additional work and thus we have only alluded to it in the discussion (Page 11, line 11-15).

[27] Sadik A, et al. IL4I1 Is a Metabolic Immune Checkpoint that Activates the AHR and Promotes Tumor Progression. *Cell*. 2020 Sep 3;182(5):1252-1270.e34. doi: 10.1016/j.cell.2020.07.038. Epub 2020 Aug 19. PMID: 32818467.

Comments: 9. The authors delineated macrophage gene expression signatures potentially indicative of survival, then prone to guide treatment decisions. These results were generated from a single-center patients selection process. In this context, the confirmation of these results on an independent validation cohort is required.

Response: We wish to reiterate that our paper does not purport to guide treatment decisions, and have carefully revised the text to clarify this. Our MacroSigs were derived from samples of reactive lymphoid tissue and DLBCL at NUH Singapore, with

the sample size limited by the high cost of such experiments (though the revised version of the study now features a high number of profiled ROI). However, we have validated the relationship of our MacroSigs with DLBCL patients' survival in the eight publicly available datasets that are from multi-center studies (spanning 4,594 patients), which comprise the validation cohorts for our findings.

Ultimately for clinical applicability, it would be valuable to exploit selected markers of the MacroSigs for IHC application to routine sections in a real world approach, however such optimisation would represent a project in itself. This would require a stepwise retrospective multiplexed study moving towards a prospective routine histopathological approach of single markers on serial sections. Nonetheless, as an example, we attempted to perform a cross-validation using the Complement classical pathway recognition molecule C1q, the transcripts of which were among the most strongly overexpressed genes in MacroSigs with negative prognostic significance (i.e. DZ and DLBCL MacroSigs, Supplementary Table 2). We validated C1q protein expression as characterizing DZ macrophages in the GC and showed that in an independent patient cohort-CMMC cohort (see Methods in main manuscript, Page 21, line 3-5; Supplementary Appendix, Page 4, line 2-18), stratification by C1q positivity evaluated in CD68+ cells associating with worse survival (Fig. 7D-E).

Minor issues:

Comments: 1. The raws are not numbered.

Response: We have added row numbers for each page.

Comments: 2. A little more effort in explaining the results should be made in the figures. The comments are too short to fully catch the results without reading the corresponding section.

Response: We thank the reviewer for highlighting this limitation in our paper. We have revised the figure legends within available space to elaborate on the results.

Comments: 3. The figures and their organization: Figure 1 and 2 for example, are made many (up to 10) (A-J) panels. A figure is something the reader should easily read to access the main information. I don't think the 10 panels of figure 2 can fit in a single printed page. Figures 6 and 7: there are too many panels.

Response: We thank the reviewer for their suggestion, and have re-arranged the figure layout to make it more concise and representative.

Comments: 4. In Figure 4 comments there are up to 8 panels (A-H). However, the corresponding figure only displays 6 panels (A-F).

Response: We apologise for this error which we have now corrected.

Comments: 5. Figure5: may the authors explain how the gene ratios were calculated?

Response: We apologise for the lack of clarity. For the current Figures 2C, 2E, 3D and Supplementary Fig. 6B, count refers to the number genes present in the overlap between the MacroSigs and the Hallmark gene sets (downloaded from Human Molecular

Signature Database [MSigDB]). The gene ratios were obtained by dividing the count by the total number of genes in that respective Hallmark gene set. For the current Fig. 4A-C, the COO association analyses are presented as dot plots, where the enrichment ratio refers to the number of patients classified as both a certain MacroSig and COO category, divided by the total number of patients classified in that particular COO category, hence the ‘gene ratio’ in these figures has now been replaced by the more appropriate term ‘enrichment ratio’. These points have now been explicitly stated in the methods section (Page 29, line 4-14).

Comments: 6. Discussion section: CCO classifications  COO classifications.

Response: We apologise for this error which we have now corrected.

Reviewer #2 (Remarks to the Author): Expert in spatial transcriptomics and tumour microenvironment

Comments: In this work, Liu et al. apply digital spatial profiling, a form of spatial transcriptomics, to diffuse large B-cell lymphoma samples, extracting gene expression data for 3 immune cell populations. They use the DSP data to identify 8 “MacroSigs” corresponding to various spatial regions and patient survival, suggesting that macrophages take on different cellular identities depending on their surroundings, and that these identities can contribute to patient outcomes.

The work is technically impressive and is a nice use case of the DSP technology, and the manuscript is written in a clear and easy to follow manner. While the biology suggested by the transcriptomics data is interesting, novel and has clear clinical relevance, the manuscript reads as a bit disjointed. The data itself is somewhat correlative/descriptive, and lacking any mechanistic experiments to validate the conclusions suggested by the transcriptomics data. The lack of corresponding scRNA-seq data makes it impossible to assess whether the MacroSigs themselves represent true macrophage cell states at the single-cell level.

In my opinion, several major changes are needed before the manuscript is suitable for publication in Nature Communications. A major weakness is the lack of detail describing the methods, particularly in how the MacroSigs were defined. This makes it difficult for me to be convinced that the MacroSigs were defined correctly and in a stringent manner, such that they truly represent novel macrophage populations. The fact that the MacroSigs not do map well onto existing macrophage scRNA-seq atlases (i.e MoMac) also raises concern that this may not in fact be due to the MacroSigs representing novel macrophage populations (as the authors suggest), but that the MacroSigs were sloppily defined (i.e too many genes, not stringent enough cutoffs). As it is likely that future researchers will use these signatures for understanding macrophage populations present in their own data, it is of critical importance to the field that (a) the MacroSigs are defined in a stringent way, and (b) the methods used to identify the MacroSigs are clearly described. Expanding both the main text and

supplementary methods sections and modifying the figures as suggested below will help to alleviate these concerns.

Response: We deeply appreciate the reviewer's insightful feedback and have taken their comments into account to enhance the clarity and rigor of our manuscript. The changes made are detailed below.

Major comments:

Comments: 1. In general, the Methods section is very light on details, making it difficult for the reader to understand how the analyses were done. This is a critical issue for such an analysis-heavy paper. While the supplementary methods have more information, both sections are fairly vaguely written (e.g. “GSEA analyses was performed with R software”). Please rewrite both the main text and supplementary methods sections with enough detail such that others are able to reproduce the results (R packages used, parameters, cutoffs, etc).

Response: We fully acknowledge that we should have been more careful in ensuring detailed methods are provided for reproducibility. We have now significantly expanded both the main text and the supplementary methods sections to provide a more comprehensive and detailed description of the methods adopted in this experimental study.

Comments: 2. Related to the above point, more information needs to be provided as to how the MacroSigs were defined - it is barely described in the text or methods. Is there any overlap between the gene lists, and if so, how is overlap accounted for? Why must genes match “either” a p-value cutoff or a log₂FC cutoff - this seems like a fairly lenient criterion (why not both a p-value and a log₂FC cutoff?). Why was a log₂FC of 0.58 chosen as the cutoff (seems low)? Why were 50 genes chosen? The supplemental methods suggest in some cases, less than 50 genes were identified - why not use a higher log₂FC cutoff or lower p-value to get cleaner lists with fewer genes, that might map better onto preexisting datasets?

Response: We thank the reviewer for their comments and apologise for the ambiguous description about the definition criteria of macrophage signatures (MacroSigs). To obtain gene lists that passed both fold change and statistical significance criteria, we increased the sample size with new additional DSP work (comprising now a total of 702 regions; 24 reactive lymphoid tissues (RLTs) and 87 DLBCL patient samples; More details in Supplementary Fig. 1A). The new MacroSigs were derived from the differentially expressed genes (DEGs) of 4 comparisons (GC versus IF; LZ versus DZ; RLT versus DLBCL; No relapse versus Relapse) based on both criteria: a. Benjamini-Hochberg adjusted *P*-value < 0.05; b. Absolute log₂ fold change ($|\log_2\text{FC}|$) > 0.58. While macrosigs1-6 fulfilled both requirements, MacroSig7 (No relapse) and MacroSig8 (Relapse) did not have DEGs after *P*-value correction. We have therefore removed MacroSig 7 and 8 from our revised manuscript, and will aim to study these with additional DLBCL cohorts in a subsequent paper. The log₂FC threshold of 0.58 (1.5 times on a non-logarithmic scale) is indeed on the lower end of typical ranges used

in existing literatures [1-2]. However, we chose a low cut-off ($|\log_2FC| > 0.58$) to be as inclusive as possible of potential changes between distinct cell numbers for each AOI using DSP [3]. We discuss this limitation of the resolution of DSP in the manuscript, and highlight that further single-cell resolved spatial transcriptomic methods may further refine distinctions of subsets of macrophages in these lymphoid regions (Page 18, line 21-22; Page 19, line 1-19).

For feature expression analysis of the MacroSigs on the MoMac-VERSE, we followed the reviewers' suggestion of testing different versions by varying the numbers of "top" genes selected. The results of reducing the number from 50 to 25 genes are shown below here (images A-D), with insufficient features in the 25 gene version for proper module scores. We have now incorporated the fold change and adjusted *P*-values for all the ranked genes of the MacroSigs in Supplementary Table 2. The table indicates that both fold change and adjusted *P*-value for the top 50 genes remain well above our defined thresholds for most of the MacroSigs. The lack of features obtained from our mapping the top 25 genes onto the MoMac-VERSE suggests that incorporating as many as 50 genes, provided these genes are of sufficient fold change and significance, can actually be advantageous for uncovering associated features through mapping onto pre-existing datasets. We therefore retained the top 50 gene version for the projection analysis (Fig. 2F-I, Fig. 3E-F in the paper). With our new MacroSigs that fulfill both fold change and adjusted *P* value thresholds, we see that the top 50 genes of MacroSig 1, 4, and 6 do overlap with distinct macrophage subclusters in the MoMac-VERSE. We have thoroughly modified the description of the MacroSigs derivation workflow in the Methods section (Page 25, line 16-22; Page 26, line 1-4) and main text of the manuscript (Page 9, line 15-20; Page 11, line 2-4) to make it clearer for readers to understand.

[1] Peart MJ, et al. Identification and functional significance of genes regulated by structurally different histone deacetylase inhibitors. *Proc Natl Acad Sci U S A*. 2005 Mar 8;102(10):3697-702. doi: 10.1073/pnas.0500369102. Epub 2005 Feb 28. PMID: 15738394; PMCID: PMC552783.

[2] Raouf A, et al. Transcriptome analysis of the normal human mammary cell commitment and differentiation process. *Cell Stem Cell*. 2008 Jul 3;3(1):109-18. doi: 10.1016/j.stem.2008.05.018. PMID: 18593563.

[3] Zimmerman SM, et al. Spatially resolved whole transcriptome profiling in human and mouse tissue using Digital Spatial Profiling. *Genome Res*. 2022 Oct;32(10):1892-1905. doi: 10.1101/gr.276206.121. Epub 2022 Sep 13. PMID: 36100434; PMCID: PMC9712633.

Figure legends: (A-D) A Module Score using the top 25 statistically significant DEGs, ranked by their log fold change, was created for each MacroSig. This score was then projected onto the uniform manifold approximation and projection (UMAP) space named “MoMac-VERSE”. Thus, the Top25 genes of each MacroSig1-6 were projected respectively onto MoMac-VERSE, respectively. However, there weren’t enough features in the MoMac-VERSE object to properly get the Module Score in the MacroSig1 (GC) and MacroSig5 (RLT). The projections of rest MacroSigs were shown.

Comments: 3. As the authors themselves suggest in the Discussion, pairing the DSP results with scRNA-seq would clarify the functional role of these macrophage populations in DLBCL, and allow for more sophisticated analyses of interactions between the different subtypes (pseudotime analysis, modelling of ligand-receptor signaling, etc). A scRNA-seq experiment would also provide critical confirmation at the single-cell level that the different MacroSigs truly represent different macrophage subtypes.

Response: We thank the reviewer for these comments and suggestions. We obtained

scRNA-seq data of B cells and Macrophage/Monocytes population of DLBCL ($N = 17$, comprising transcriptomes of 94,324 cells) from Ye et al. [4] to evaluate whether indeed our MacroSigs correspond to Macrophage/Monocytes subsets using a module score. As expected, most MacroSigs (except for GC/ RLT which share several cell-cycle/proliferation genes with the GC/RLT B-cell signature from our DSP experiment) showed a clear enrichment and higher module scores in the Monocytes/Macrophages population over the B-cell population (Fig. 4D and Supplementary Fig. 5). These results suggest that the MacroSigs do identify macrophage subtypes within DLBCL samples. Further subcategorisation of macrophages and sophisticated analyses was limited by the number of monocyte-macrophage cells in this dataset [5], reflective of the general limitation of scRNA-seq in that macrophages are often underrepresented. We hope to in the future perform single-cell resolved spatial transcriptomic technique to further explore these aspects.

[4] Ye X, et al. A single-cell atlas of diffuse large B cell lymphoma. *Cell Rep.* 2022 Apr 19;39(3):110713. doi: 10.1016/j.celrep.2022.110713. PMID: 35443163.

[5] Kolodziejczyk AA, et al. The technology and biology of single-cell RNA sequencing. *Mol Cell.* 2015 May 21;58(4):610-20. doi: 10.1016/j.molcel.2015.04.005. PMID: 26000846.

Comments: 4. The description of the data presented in Fig. 5 needs improvement and to be expanded - there is a huge amount of biology presented here that is almost glossed over, with passing references to many different tumor subtypes and cell states that a non-DLBCL expert will have no knowledge of. Furthermore, the survival data, while mentioned in the text, is not shown in the figure. I am not sure if so many dot plots are the best way to display this data. Perhaps the data could be plotted another way in which the corresponding survival data is also shown.

Response: We thank the Reviewer for this valuable feedback. We have redone the original Fig. 5 (now labeled as Fig. 4). For Fig.4A-C about the conventional DLBCL COO subtypes, we stick to dotplots which summarise the relationships of our spatially-derived MacroSigs with DLBCL COO. This figure does not aim to discuss relationships with survival, so we have removed the corresponding sentence in the text. For previous Fig. 5D-F about the genetic and molecular DLBCL subtypes, we have combined the results of three datasets (Refs 6-8 below) and changed the dotplots into bar graphs based on strength of enrichment [6-8] and displayed it in the Supplementary Fig. 6A. However, there are no clear associations that are consistent between the three studies. We retain the dotplot to depict the lack of relationship of our MacroSigs with DLBCL TME categories proposed by Kotlov et al [9] in Supplementary Fig. 6B.

[6] Schmitz R, et al. Genetics and Pathogenesis of Diffuse Large B-Cell Lymphoma. *N Engl J Med.* 2018 Apr 12;378(15):1396-1407. doi: 10.1056/NEJMoa1801445. PMID: 29641966; PMCID: PMC6010183.

[7] Lacy SE, et al. Targeted sequencing in DLBCL, molecular subtypes, and outcomes: a Haematological Malignancy Research Network report. *Blood.* 2020 May

14;135(20):1759-1771. doi: 10.1182/blood.2019003535. PMID: 32187361; PMCID: PMC7259825.

[8] Chapuy B, et al. Molecular subtypes of diffuse large B cell lymphoma are associated with distinct pathogenic mechanisms and outcomes. *Nat Med.* 2018 May;24(5):679-690. doi: 10.1038/s41591-018-0016-8. Epub 2018 Apr 30. Erratum in: *Nat Med.* 2018 Aug;24(8):1292. Erratum in: *Nat Med.* 2018 Aug;24(8):1290-1291. PMID: 29713087; PMCID: PMC6613387.

[9] Kotlov N, et al. Clinical and Biological Subtypes of B-cell Lymphoma Revealed by Microenvironmental Signatures. *Cancer Discov.* 2021 Jun;11(6):1468-1489. doi: 10.1158/2159-8290.CD-20-0839. Epub 2021 Feb 4. PMID: 33541860; PMCID: PMC8178179.

Minor comments:

Comments: 1. Top of page 6: please describe which cell types CD68, CD3, and CD20 are labelling - it is written in the figure legend but not in the manuscript text. In general, it would be helpful for the reader if the corresponding cell type is mentioned every time a cell type marker is referred to.

Response: We thank the Reviewer for this comment. In the revised version of the figures we have aimed to emphasise the correspondence between IF markers and the cell types.

Comments: 2. Can the authors comment on the approximate size of each ROI and AOI? Approximately how many cells are captured per AOI? This is important to understand the resolution of the DSP data.

Response: A table summarising the size and cell numbers per AOI has been included in the Supplementary table 6.

Comments: 3. Please change the colours in Fig. 1B, bottom right panel - the 2 purple colours labelling the CD68 and CD20 populations are difficult to distinguish.

Response: We apologise that we are unable to change this for technical reasons. Unfortunately, the colours of each mask can only be assigned during the ROI selection process of the experiment on the DSP instrument itself, and these were the colours chosen at the time. We will take note of this for subsequent manuscripts.

Comments: 4. Fig. S2B: I am curious why the clustering of the DLBCL scRNA-seq data is so poor, with many overlapping populations that I would expect to be fairly separate in UMAP space based on gene expression. Can the authors comment on how they integrated these 10 different datasets? Including a UMAP in which the cells are labelled by original dataset would also be helpful to visualize how well the integration worked.

Response: We thank the reviewer for this comment. The new analysis has now been performed on a single larger dataset of scRNA-seq in DLBCL (N = 17, comprising transcriptomes of 94,324 cells) obtained from Ye et al. [4]. The original cell annotations can be found within Ye et al [4] and we have referenced this in the manuscript.

[4] Ye X, et al. A single-cell atlas of diffuse large B cell lymphoma. Cell Rep. 2022 Apr 19;39(3):110713. doi: 10.1016/j.celrep.2022.110713. PMID: 35443163.

Comments: 5. Fig. S2C,D: Diverging colour maps should not be used for unscaled data.

Response: We thank the reviewer for this suggestion, and have reconfigured Supplementary Fig. 2 and split the panels across Supplementary Fig. 2-3.

Comments: 6. Fig. 2: how well can the authors define MacroSigs corresponding to LZ vs DZ when there are no statistically significant DEGs between the two groups? Same goes for Fig. 4B.

Response: To address this issue, we increased the sample size with a new additional DSP experiment (A total of 702 regions; 24 reactive lymphoid tissues (RLTs) and 87 DLBCL patients' samples; More details are provided in Supplementary Fig. 1A). In the revised version of the manuscript, we derived the MacroSigs³⁻⁴ (LZ/DZ) from DEGs based on two criteria: a. Benjamini-Hochberg adjusted P -value < 0.05 ; b. Absolute \log_2 fold change ($|\log_2FC| > 0.58$). (Page 9, line 15-20)

Comments: 7. Figs. 2G-J and 3E-H: I am not sure what purpose this analysis serves, other than indicating that the MacroSigs poorly correspond to different macrophage subtypes. 50 genes are a lot to be plotting, so I'm not surprised that the authors see poor cell-type specific expression of the MacroSigs. This goes back to the issue of how the MacroSigs were defined - with so few details, it is hard for the reader to be convinced that the MacroSigs are a stringent representation of these cell populations.

Response: We duly noted this comment and apologise for the lack of details in the previous version of the manuscript. We have thoroughly modified the writing about the derivation of MacroSigs in the Methods section (Page 25, line 16-22; Page 26, line 1-4) and main text of the manuscript (Page 9, line 15-20; Page 11, line 2-4). In the revised manuscript, all MacroSigs were generated in a more stringent and robust manner (thanks to the extended DSP dataset). As mentioned above, there were insufficient features in the 25 gene version for proper module scores and we therefore retained the top 50 gene version for the projection analysis (Fig. 2F-I, Fig. 3E-F in the paper). With our new MacroSigs that fulfill both fold change and adjusted p -value thresholds, we see that the top 50 genes of MacroSig 1,3,4, and 6 do overlap with distinct macrophage subclusters in the MoMac-VERSE.

Comments: 8. I am not sure if Fig. 3A is needed - it is not very convincing (shows little difference in gene expression between the DLBCL and RLT AOIs) and the clustering is not great. Presumably the authors know which AOIs came from which tissue without the need for this analysis. Fig. 3C is much more convincing in showing changes in gene expression between the two groups.

Response: We have removed the original Figure 3A following the reviewers' suggestion.

Comments: 9. Fig. 3: it would be helpful to include a heatmap with the expression of the genes corresponding to MacroSigs 5 and 6 plotted for DLBCL and RLT AOIs.

Response: Following the Reviewer's suggestion, we have made a new heatmap that displaying the top 25 genes highly expressed in the RLT and DLBCL samples (new Fig. 3C).

Comments: 10. Fig. 5: It would be interesting to see how the MacroSigs map in UMAP space for the datasets in Fig. 5E-H (if these are in fact single cell datasets - this is not mentioned in the figure legend or the text).

Response: We thank the reviewer for pointing this out. The datasets used for Fig. 5E-H are bulk RNA-seq datasets so we could not project them into the UMAP. We have made more efforts to revise the figure legend and the main text for clarity (Page 11, line 20-22).

Comments: 11. Fig. 6: The authors map the MacroSigs onto bulk RNA-seq data from DLBCL patient samples, and state that the RNA-seq data likely contains macrophages since they are such a large component of the TME. What proportion of macrophages (relative to tumor) are found in these datasets? Can the authors infer the approximate proportion of macrophages based on macrophage gene expression in the bulk RNA-seq data?

Response: We would like to thank the Reviewer for this insightful comment. We have attempted to evaluate the macrophage proportion in the RNA bulk datasets through a consolidated immune deconvolution approach [10]. It is likely that the reads of genes from our MacroSigs within these DLBCL bulk RNA datasets are representative of macrophage infiltration. Accordingly, we demonstrate non-0 counts for macrophage presence in most cases using an immune deconvolution algorithm. This analysis has been provided for the Reviewer's perusal as Supplementary Data.

[10] Aran D, Hu Z, Butte AJ. xCell: digitally portraying the tissue cellular heterogeneity landscape. *Genome Biol.* 2017 Nov 15;18(1):220. doi: 10.1186/s13059-017-1349-1. PMID: 29141660; PMCID: PMC5688663.

Comments: 12. The Discussion is too long (almost 5 pages) and should be cut down.

Response: We have condensed the Discussion section for brevity and clarity.

Comments: 13. Supplementary tables should be included as Excel or CSV files - not PDFs.

Response: We thank the reviewer for highlighting this. We uploaded the Supplementary tables as csv files into the journal portal. It may have been automatically converted into PDF files for the review process. We have highlighted this to the editor and will aim to provide the tables in csv format.

Comments: 14. The paper relies heavily on acronyms - it seems like overkill at times

and makes it difficult to read. Page 12 itself must contain close to 20 different acronyms.

Response: We thank the reviewer for highlighting this. We have edited the manuscript to reduce the number of acronyms used wherever possible.

Reviewer #3 (Remarks to the Author): Expert in lymphomas, immunology, and macrophages

In this manuscript, entitled "Spatially-resolved transcriptomics reveal macrophage heterogeneity and prognostic significance in diffuse large B-cell lymphoma", Liu et al, characterized macrophages in DLBCL tissues by using a spatial transcriptome approach.

Recent single-cell RNA-sequencing studies have revealed heterogeneity of tumor-associated macrophages; however, spatial heterogeneity remains to be fully characterized. This manuscript defines macrophages in DLBCL tissues by a spatial transcriptomics approach. Specifically, the authors defined 8 spatially-derived macrophage signatures (MacroSigs), which could clearly separate macrophages within normal reactive lymph nodes, as well as macrophages from different disease statuses (normal vs DLBCL vs relapsed DLBCL). Overall, the authors carefully analyzed data, and results in this manuscript contain high novelty and significance.

I have a few suggestions.

Comments: # 1. The authors defined eight spatially-derived macrophage signatures. To validate the results, the authors investigated clinical relevance (Cell-of-Origin and prognosis), using published DLBCL datasets. However, the authors need to validate key macrophage signatures by IHC staining. This would provide more clinical implications (i.e., clinic-pathological analysis in the real-world clinical setting).

Response: We thank the reviewer for their suggestion, and agree this would be a valuable strategy for the future. We looked into the possibility of establishing a multiplex IHC panel to cross-validate our findings. However, choosing 3-5 markers from a signature with expression of over 100 transcripts is not straightforward (for example none of the IHC algorithms for COO perform as well as the RNA signature). In this paper we focus on the DZ MacroSig in view of its strong correlation with survival. As C1q is highly expressed in DZ- and DLBCL macrophages (Supplementary Table 2), and C1q expressing macrophages are a defined physiological entity, we evaluated C1q expression by multiplexed IHC. We see that C1q macrophages are indeed enriched in the dark zone (Fig. 7C) and that expression of C1q within CD68 cells stratifies patients for survival in an independent CMMC cohort of DLBCL (Fig. 7D-E). These data are now included in the revised manuscript.

Comments: #2. In addition to CD163, could the authors see differences in the expression levels of commonly used M2-related markers among 8 MacroSigs (such as

Arg-1, CD204, CD206, VEGF)?

Response: We thank the reviewer for suggesting this interesting analysis. We evaluated but did not see enrichment of these canonical M2 markers in our MacroSigs. Among other reported M2 markers, we note CD209 expression in the DZ-macrophages compared to the LZ-macrophages, while CSF1R, IL10, and TGF β -associated genes (TGFB1, TGFB1) are highly expressed in the DLBCL-macrophages compared to the RLT-macrophages. We have alluded to this in the discussion (Page 17, line 12-15).

Comments: #3. It is now recognized that macrophage phagocytosis checkpoint molecules are associated with resistance to rituximab. Could the authors see differential expression levels of molecules (SIRPa, Siglec-10, LILRB1, and PD-1) among 8 MacroSigs?

Response: We thank the reviewer again for suggesting this interesting analysis. We have now checked but do not see a differential expression of SIRPa, Siglec-10, LILRB1, and PD-1 in our six MacroSigs themselves. Interestingly, we do see that these macrophage checkpoints are enriched in patients categorised to have MacroSig 6 (DLBCL-like, in comparison to MacroSig 5- RLT like) in our GEP analysis (Supplementary Fig. 7). We have alluded to these findings in the revised manuscript (Page 17, line 15-22).

REVIEWERS' COMMENTS

Reviewer #2 (Remarks to the Author):

The manuscript by Liu et al. is much improved - I especially appreciate the authors' efforts to clarify the Methods section and their work in adding more DSP samples and single cell analyses. I am glad to see using more stringent cutoffs in defining the MacroSigs has resulted in cleaner results and better overlap with existing data. I have a few minor suggestions to help improve readability of the manuscript and interpretation of the data.

1. The plots in Fig. 2 are so small as to be almost unreadable (not sure if this was an issue with how the PDF was rendered). The heatmap in particular is tiny with what looks like completely overlapping gene names on the right side. Please reorganize to make more legible. Same goes for Fig. 3A-C and many of the supplemental figures which are very difficult to read, particularly S5. I also would probably label fewer genes on Fig. 2B and 2D to be able to increase the size of the labels on the remaining genes.
2. Please add the UMAP with cell type annotations for the MoMAC-verse data as a panel in Fig. 2 - i.e., make the small multicoloured UMAP at the top right of Fig. 2F-I its own panel with all macrophage subtypes annotated in a legend. This will make it much easier for the reader to interpret the results in 2F-I and 3E-F, especially in the cases where the gene modules do not map to one specific population. For example, MacroSig3 does appear to be enriched in a specific population but it is not labeled in the UMAP at the top right or mentioned in the text.
3. Not sure if Fig. 3E is necessary since it's not referred to in the text - could be removed to make space to make the other panels bigger (to help with legibility issues mentioned above).
4. I appreciate the increased details in the Methods section. However, there are still some small areas that would benefit from clarification:
 - a. In the Datasets subsection, please clarify what the Ns refer to for each dataset - Patients? Samples? Cells?
 - b. Within the Datasets section there is unnecessary information that, if anything, should be in the Acknowledgements (e.g. "The Genomic Variation in DLBCL study was supported by...")
5. This might be semantics but I think the authors are using "GSEA" as a catchall term encompassing both pathway analysis (overlap between their gene lists and existing pathway sets using the hypergeometric test) and traditional GSEA (calculating an enrichment score based on \log_2FC), which is incorrect as these are two distinct analyses. They also switch between referring to "gene ratio" and "enrichment ratio" in for example Fig. 3D and Fig. 4 when it's not clear if they are referring to the results from pathway analysis or GSEA. Please clarify in each case if pathway analysis or GSEA is being performed and standardize labels across plots.

Reviewer #3 (Remarks to the Author):The authors have addressed my comments. The quality has been significantly improved in this revised manuscript.

Reviewer #4 (Remarks to the Author):

The authors addressed most of the comments of referee #1 adequately and improved the manuscript considerably. A major advancement is the significantly increased cohort size, including also a better description of the sample selection, which allowed for an improved statistical analysis.

However, a few open questions remain:

Comment 3d: It remains unclear how large the AOIs are? Suppl. Table 6 gives numbers but no units to the size. The IF image in Fig. 1 shows single CD68+ cells in the LN sections, but Suppl. Table defines the content of each CD68+ AOI of in average 80 cells, with up to several hundred cells per AOI. It is unclear how the transcriptome data of the AOI can therefore be specific for CD68+ cells. This needs to be better described and explained.

To comment 4d: The authors did not identify a clear annotation of the MacroSigs using the MoMac-Verse annotations. This might be due to not having single-cell resolved data from the GeoMx platform. The authors should include this in the discussion.

To comment 8: As the authors identified the DLBCL-specific gene signature MacroSig6 to be similar to the MoMac-Verse population of IL4i1+ macrophages, they speculate that these macrophages might be targets of AhR inhibitors, as a therapeutic approach. But looking at the DEG of DLBCL macrophages (MacroSig6), IL4i1 is not among the list. Do the authors still consider is a viable approach and why? This has to be further explained.

We thank the reviewers for their time to read through our manuscript. The comments and suggestions have significantly improved our work quality. We are very appreciative of their constructive insight and have revised the manuscript based on it. We hope the revised version and the point-by-point replies below (in blue) will address their concerns.

Responses to reviewer comments:

Reviewer's Comments:

Reviewer #2 (Remarks to the Author)

The manuscript by Liu et al. is much improved - I especially appreciate the authors' efforts to clarify the Methods section and their work in adding more DSP samples and single cell analyses. I am glad to see using more stringent cutoffs in defining the MacroSigs has resulted in cleaner results and better overlap with existing data. I have a few minor suggestions to help improve readability of the manuscript and interpretation of the data.

Response: We deeply appreciate the reviewers' insightful feedback and have taken the comments into account to enhance the readability of our manuscript. The changes made are detailed below.

1. The plots in Fig. 2 are so small as to be almost unreadable (not sure if this was an issue with how the PDF was rendered). The heatmap in particular is tiny with what looks like completely overlapping gene names on the right side. Please reorganize to make more legible. Same goes for Fig. 3A-C and many of the supplemental figures which are very difficult to read, particularly S5. I also would probably label fewer genes on Fig. 2B and 2D to be able to increase the size of the labels on the remaining genes.

Response: We thank the reviewer for their suggestions, and have re-arranged the Fig. 2, Fig. 3, and Supplementary Fig. 5 to make it clear and readable. Specifically, we have reduced the number of genes labelled on the volcano plots, highlighting only selected genes and increasing the label size.

2. Please add the UMAP with cell type annotations for the MoMAC-verse data as a panel in Fig. 2 - i.e, make the small multicoloured UMAP at the top right of Fig. 2F-I its own panel with all macrophage subtypes annotated in a legend. This will make it much easier for the reader to interpret the results in 2F-I and 3E-F, especially in the cases where the gene modules do not map to one specific population. For example, MacroSig3 does appear to be enriched in a specific population but it is not labeled in the UMAP at the top right or mentioned in the text.

Response: We have added the cluster annotation for the MoMac-VERSE as a panel in Fig. 2F, and re-configured Fig.2 and Fig.3 as suggested. Thank you for pointing out the absence of labelling for MacroSig3, we have now incorporated this into Fig. 2F and within the manuscript.

3. Not sure if Fig. 3E is necessary since it's not referred to in the text - could be removed to make space to make the other panels bigger (to help with legibility issues mentioned above).

Response: We have deleted the Fig. 3E and re-arranged the Fig. 3 to make it clear and readable.

4. I appreciate the increased details in the Methods section. However, there are still some small areas that would benefit from clarification:

a. In the Datasets subsection, please clarify what the Ns refer to for each dataset - Patients? Samples? Cells?

Response: We have added a sentence to clarify all “n” are patient numbers in the Data availability section. (Page 30, line 19-20)

b. Within the Datasets section there is unnecessary information that, if anything, should be in the Acknowledgements (e.g “The Genomic Variation in DLBCL study was supported by...”)

Response: We have moved these sentences to the Acknowledgements. (Page 36, line 36-37-Page 37, line 1-2)

5. This might be semantics but I think the authors are using “GSEA” as a catchall term encompassing both pathway analysis (overlap between their gene lists and existing pathway sets using the hypergeometric test) and traditional GSEA (calculating an enrichment score based on log₂FC), which is incorrect as these are two distinct analyses. They also switch between referring to “gene ratio” and “enrichment ratio” in for example Fig. 3D and Fig. 4 when it's not clear if they are referring to the results from pathway analysis or GSEA. Please clarify in each case if pathway analysis or GSEA is being performed and standardize labels across plots.

Response: We apologise for the inaccurate terminology, indeed ‘GSEA analysis’ should be more correctly rendered ‘pathway enrichment analysis’; we have amended this within the text.

We also apologise for the lack of clarity in terminology between ‘gene ratio’ and “enrichment ratio”. For the pathway analysis, we have used the term ‘gene ratio’, to describe the value obtained by dividing the ‘count’ by the total number of genes in a respective Hallmark gene set. The term ‘gene ratio’ is similar but distinct from the ‘enrichment ratio’ that was used for our association analyses. The latter ratio refers to, for example, the number of patients classified as both a certain MacroSig and COO category, divided by the total number of patients classified in that particular COO category. For clarity, within the text and figures we have renamed ‘enrichment ratio’ as ‘overlap ratio’.

Reviewer #3 (Remarks to the Author)

The authors have addressed my comments. The quality has been significantly improved

in this revised manuscript.

Response: We deeply appreciate the reviewer's insightful feedback and have markedly improved the quality of our manuscript.

Reviewer #4 (Remarks to the Author):

The authors addressed most of the comments of referee #1 adequately and improved the manuscript considerably. A major advancement is the significantly increased cohort size, including also a better description of the sample selection, which allowed for an improved statistical analysis.

However, a few open questions remain:

Comment 3d: It remains unclear how large the AOIs are? Suppl. Table 6 gives numbers but no units to the size. The IF image in Fig. 1 shows single CD68+ cells in the LN sections, but Suppl. Table defines the content of each CD68+ AOI of in average 80 cells, with up to several hundred cells per AOI. It is unclear how the transcriptome data of the AOI can therefore be specific for CD68+ cells. This needs to be better described and explained.

Response: We showed the AOI size in the column named “Area” of Supplementary Data 3- this has now been relabelled as AOI size (“ μm^2 ”) for clarity;

The specificity of an AOI for a given cell type (eg CD68+ cells) is based on the DSP methodology, where each AOI- defined by the cellular masks (e.g. CD68) within a given ROI- is exposed to UV light for photocleaved oligos to be aspirated from the solution into the wells of a collection plate for downstream sequencing and data processing. We therefore obtain cell-type specific transcriptome data for all cells marked by the mask, across a range of AOI sizes. The gene counts obtained from each AOI are the aggregate from all cells of a given cell type within an ROI (normalised as described in Methods: DSP data processing and harmonization). These details have been elaborated on in the methods section. (Page 23, line 1-14)

To comment 4d: The authors did not identify a clear annotation of the MacroSigs using the MoMac-Verse annotations. This might be due to not having single-cell resolved data from the GeoMx platform. The authors should include this in the discussion.

Response: Yes this is indeed a limitation of the method, and we have added it in the Discussion section. (Page 16, line 8-10)

To comment 8: As the authors identified the DLBCL-specific gene signature MacroSig6 to be similar to the MoMac-Verse population of IL4i1+ macrophages, they speculate that these macrophages might be targets of AhR inhibitors, as a therapeutic approach. But looking at the DEG of DLBCL macrophages (MacroSig6), IL4i1 is not among the list. Do the authors still consider is a viable approach and why? This has to be further explained.

Response: MacroSig6 could still represent a population of IL4I1+ macrophages despite the eponymous gene itself not appearing in the signature. It is possible that the

transcription of the gene and translation and storage of the secretory product, IL4I1, may not be strongly correlated in real time, in other words, when a macrophage is denoted IL4I1+ by virtue of its store of IL4I1 in its secretory vesicles, transcriptional activity of this gene may no longer be present.

As IL4I1-mediated tryptophan metabolites (indole-3-pyruvic acid, etc) are the key metabolite mediating IL4I1-driven AHR Activity in malignancy (e.g. promoting tumor cell motility and suppression of T cell responses) [1-2], AhR inhibitors may still be a possible strategy of interest We highlight that additional mechanistic work will be needed to confirm this.

[1] Sadik A, et al. IL4I1 Is a Metabolic Immune Checkpoint that Activates the AHR and Promotes Tumor Progression. *Cell*. 2020 Sep 3;182(5):1252-1270.e34. doi: 10.1016/j.cell.2020.07.038. Epub 2020 Aug 19. PMID: 32818467.

[2] Zeitler L, et al. Anti-ferroptotic mechanism of IL4i1-mediated amino acid metabolism. *Elife*. 2021 Mar 1;10:e64806. doi: 10.7554/eLife.64806. PMID: 33646117; PMCID: PMC7946422.